# Deciphering the state of immune silence in fatal COVID-19 patients

Pierre Bost[1,2,3,11], Francesco De Sanctis[4,11], Stefania Canè[4,11], Stefano Ugel [4], Katia Donadello[5], Monica Castellucci[6], David Eyal[1], Alessandra Fiore[4], Cristina Anselmi[4], Roza Maria Barouni[4], Rosalinda Trovato[4], Simone Caligola [4], Alessia Lamolinara [7], Manuela Iezzi [7], Federica Facciotti [8], Annarita Mazzariol[9], Davide Gibellini[9], Pasquale De Nardo[10], Evelina Tacconelli[10], Leonardo Gottin[5], Enrico Polati[5], Benno Schwikowski [2], Ido Amit [1✉] & Vincenzo Bronte [4✉]

Since the beginning of the SARS-CoV-2 pandemic, COVID-19 appeared as a unique disease with unconventional tissue and systemic immune features. Here we show a COVID-19 immune signature associated with severity by integrating single-cell RNA-seq analysis from blood samples and broncho-alveolar lavage fluids with clinical, immunological and functional ex vivo data. This signature is characterized by lung accumulation of naïve lymphoid cells associated with a systemic expansion and activation of myeloid cells. Myeloid-driven immune suppression is a hallmark of COVID-19 evolution, highlighting arginase-1 expression with immune regulatory features of monocytes. Monocyte-dependent and neutrophil-dependent immune suppression loss is associated with fatal clinical outcome in severe patients. Additionally, our analysis shows a lung CXCR6[+] effector memory T cell subset is associated with better prognosis in patients with severe COVID-19. In summary, COVID-19-induced myeloid dysregulation and lymphoid impairment establish a condition of 'immune silence' in patients with critical COVID-19.

[1] Department of Immunology, Weizmann Institute of Science, Rehovot, Israel. [2] Systems Biology Group, Department of Computational Biology and USR 3756, Institut Pasteur and CNRS, Paris, France. [3] Sorbonne Universite, Complexite du vivant, Paris, France. [4] Immunology Section, Department of Medicine, University and Hospital Trust of Verona, Verona, Italy. [5] Intensive Care Unit, Department of Surgery, Dentistry, Maternity and Infant, University and Hospital Trust of Verona, Verona, Italy. [6] The Center for Technological Platforms, University of Verona, Verona, Italy. [7] CAST- Center for Advanced Studies and Technology, Department of Neurosciences, Imaging and Clinical Sciences, University of G. D'Annunzio of Chieti-Pescara, Chieti, Italy. [8] Department of Experimental Oncology, IEO European Institute of Oncology IRCCS, Milan, Italy. [9] Microbiology Unit, Department of Diagnostics and Public Health, University and Hospital Trust of Verona, Verona, Italy. [10] Division of Infectious Diseases, Department of Diagnostics and Public Health, University and Hospital Trust of Verona, Verona, Italy. [11]These authors contributed equally: Pierre Bost, Francesco De Sanctis, Stefania Canè. ✉email: ido.amit@weizmann.ac.il; vincenzo.bronte@univr.it

Severe acute respiratory syndrome coronavirus 2 (SARS-CoV-2) is the etiological agent of the coronavirus disease (COVID-19) outbreak[1] that is currently threatening worldwide health. Italy was the first European nation to be severely affected by COVID-19: the first death was reported on 21 February 2020[2]; and as of 21 July 2020 more than 35,000 and 607,781 COVID-19-related deaths were registered in Italy and worldwide, respectively.

Many studies highlight different, stepwise patterns of disease progression, characterized by mild-to-moderate features in most of the patients, with some of them who unfortunately progress to a more-severe disease stage, which can lead to acute respiratory distress syndrome (ARDS), respiratory failure, and eventually death[3,4]. The contribution of host immune system in establishing the worse prognosis has been already confirmed by several clinical observations on SARS-CoV-2 and other SARSs-dependent diseases. Indeed, lymphopenia and release of pro-inflammatory cytokines such as CXCL10 (IP-10), interleukin (IL)-6, IL-8, IL-10, tumor necrosis factor (TNF) and C-C motif chemokine ligand (CCL)2 are enlisted as hallmark of severe SARS-CoV2 infection and correlate with adverse clinical outcome[4–6]. Accordingly, multicenter analysis on hospitalized COVID-19 patients established among clinical parameters associated to critical outcome not only age, co-morbidities and pre-existing diseases but also immune alterations such as an increased neutrophil to lymphocyte ratio[7], hinting that pathogenic disease characteristics are worsened in sub-optimally efficient and immune dysfunctional patients.

Whether a dysregulated host immune system, characterized by the coexistence between pro-inflammatory and anti-inflammatory mediators, represents a key feature of COVID-19 severe progression, as well as the molecular mechanisms driving this imbalance, are not elucidated yet. Indeed, ARDS experienced by COVID-19 patients has a unique signature that differs from ARDS caused by any other infective or traumatic insults[8]. More specifically, the increase in cytokine release in peripheral blood, often associated with disease severity[9] and commonly defined as "cytokine storm", is only partially involved in COVID-19 patients. Indeed, IL-6 plasma levels in COVID-19 severe patients are 10- to 40-fold lower than previously reported ARDS patients, and 1000-fold lower compared with patients facing cytokine release syndrome following treatment with chimeric antigen receptor T cells[10,11]. Thus, it is conceivable that SARS-CoV-2 infection may hijack host immune system in order to impair antiviral immunity and trigger a chronic inflammation characterized, but not limited, by the accumulation of inflammatory cytokines that participate in acute lung injury in severe COVID-19 patients. Recent literature explored the ability of CoVs to skew cytokine release by affecting IFN-dependent, antiviral response towards other inflammatory pathways sustaining the activation of inflammasome[12]. Noteworthy, delayed type I IFN signaling impairs antigen-specific T-cell responses and promotes high cytokine secretion in lung by incoming monocytes, resulting in vascular leakage and fatal disease in SARS-CoV-infected mice[13]; furthermore, type I IFN, T cells, and signal transducer and activator of transcription 1 are required for virus clearance and disease resolution in a mouse model of SARS-Cov2 infection[14] and impaired type I IFN activity results in worse outcome in human COVID-19-infected patients[15,16].

Severe COVID-19 patients display some shared features of sepsis, including secretion of inflammatory cytokines, neutrophil activation, reduced function of natural killer cells (NK), and dendritic cells (DC), altered monocyte activation, and lymphopenia[6,17]. Several high-dimensional phenotypic and molecular approaches were deployed in order to dissect the biology of virus–immune system interaction during COVID-19 pathogenesis[6,15,18–22]. These analyses were performed on peripheral blood and peripheral blood mononuclear cells (PBMCs) isolated from patients with different disease severity. This strategy streamlines sample accessibility, identification of new peripheral predictive biomarkers, and allows comparison within different studies with the caveat of not considering the local microenvironment in which the virus is acting. Taking together, all these studies highlight the presence of an IFN signature in mild-to-moderate patients and evidence a sustained emergency myelopoiesis associated with an increase in immature neutrophils and monocytes with immunesuppressive features in critically ill patients. Unfortunately, none of the published studies analyzed the immune regulatory properties of myeloid cells at the functional level. A specific genetic locus including immune-related genes (such as C-X-C motif chemokine receptor 6—CXCR6), was found to be associated with worse prognosis in COVID-19 patients[23]. Studies exclusively performed on bronchoalveolar lavage fluids (BAL), elucidated the sustained interplay between macrophages releasing inflammatory cytokines and lung epithelial cells in more-severe COVID-19 disease stages[21], whereas others pointed to clonally expanded CD8+ T cells in moderate patients[22].

Here we provide an atlas of the immune landscape of COVID-19 patients, integrating molecular (single-cell RNA sequencing, scRNA-seq), functional and clinical data from local (lung) and systemic (blood) tissues in order to define the complex interplay between SARS-CoV-2 and the host immune system. In patients with severe COVID-19, we show innate and adaptive immune dysfunction, including loss of immunesuppression by blood myeloid cells and the replacement of lung memory CD8+ T cells by naive T cells, suggesting a state of "immune silence" that correlates with a severe clinical manifestation and fatal outcome.

## Results

### Establishment of BAL and blood immune cell atlas of COVID-19 patients.

To gain insights into the immune deviation induced by SARS-CoV-2 virus in COVID-19 patients, we performed scRNA-seq analysis on BAL and matched peripheral blood samples obtained from 21 severe patients admitted to Intensive Care Units (ICU) and on peripheral blood of six mild SARS-CoV-2-positive patients and five healthy donors (Fig. 1a). Immunological features were assessed on the same cohorts, integrating four more mild SARS-CoV-2 patients, by multiplex enzyme-linked immunosorbent assay (ELISA), multiparametric flow cytometry, and functional assay (see Methods). All the patients were hospitalized at the University Hospital Integrated Trust of Verona. Anagraphic information and clinical characteristics of enrolled patients and healthy donors are summarized in Tables 1 and 2, respectively.

Following strict quality controls (Supplementary Figure 1a–c), cells were analyzed using the Pagoda2 pipeline[24] and clustered using Leiden community detection method[25]. The number of analyzed high-quality cells was comparable between groups (Supplementary Figure 1d). Fourteen significant cell clusters (>1% of the cells) were identified, and gathered into four major cellular subsets based on their mean expression profiles (Fig. 1b, c, Supplementary Figure 1e): epithelial cells, lymphocytes, monocytes–macrophages, and neutrophils. The epithelial cell compartment contained only one cell cluster, which was characterized by the expression of WFDC2, SPLI, and keratin genes, e.g., KRT8 and KRT19. 5. Different lymphocyte clusters were identified, namely B cells (CD79A, CD74), NK cells (CD247, GZMB, GNLY), CD8+ T cells (GZMA, CD8A), CD4+ T cells (LTB), and gamma-delta (γδ)-T cells (GZMH). Absolute cell number of those subsets, for each patient and tissue origin, are shown in supplementary table 1. Three main clusters of the monocyte–macrophage compartment were depicted in our data

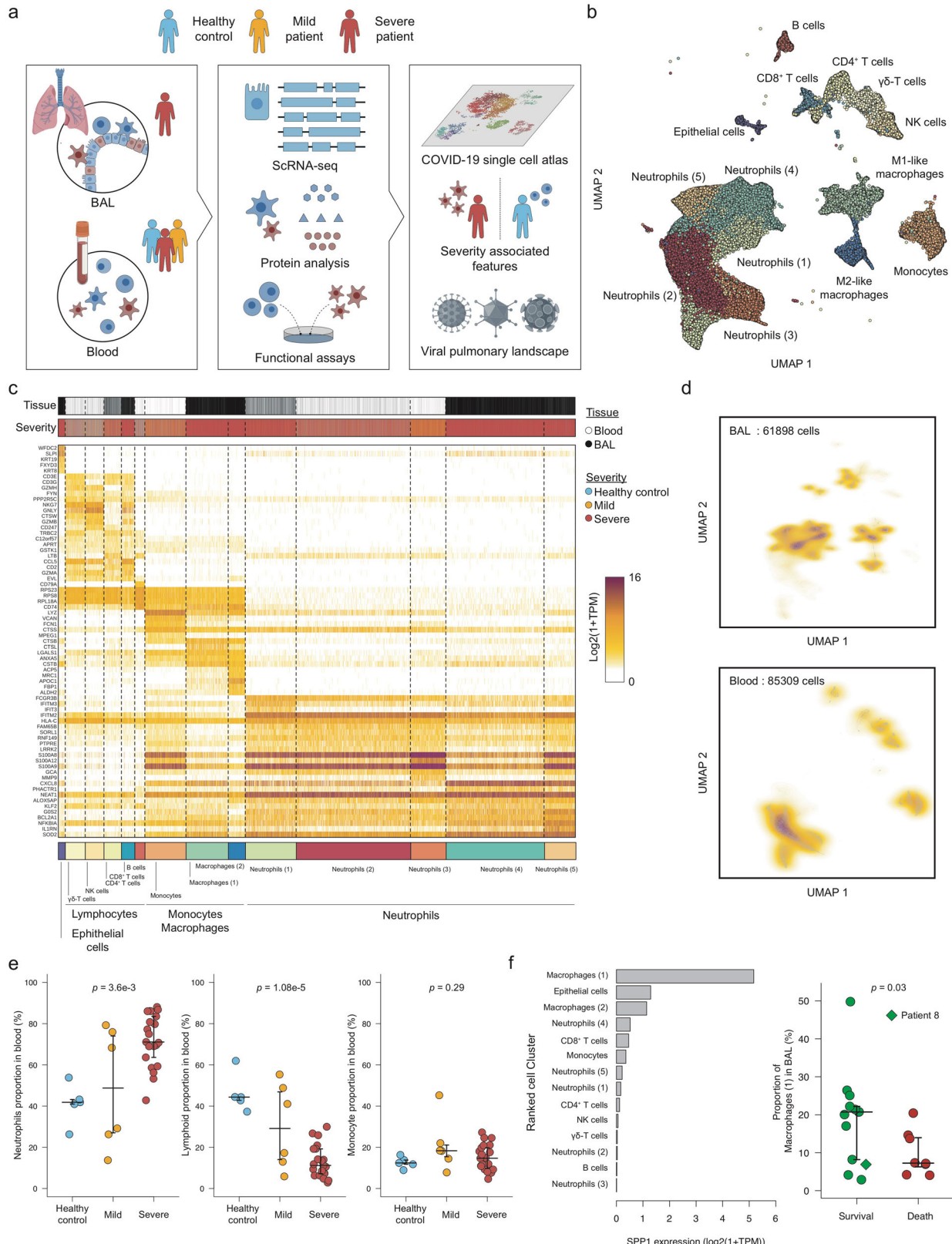

set, monocytes (*LYZ, VCAN, FCN1*), and two different macrophages clusters, the first one expressing high levels of cathepsin genes such as *CTSB* and *CTSL* (Macrophage (1)), and the second expressing metabolic-related genes (*ACP5, FBP1, ALDH2*) and named macrophage (2). At last, diversity of the neutrophil compartment was observed with five main subsets

(Supplementary Data 1). Those clusters could be further differentiated based on the expression of key markers such as *CD16B* (*FCGR3B*) mostly expressed in neutrophil clusters (1), (2), and (3), interferon response genes and Fc receptor (*IFITM3, IFIT3,* and *FCGR3B*) in cluster (1), S100 calcium-binding proteins (*S100A8, S100A9,* and *S100A12*) in clusters (3) and (5), *CXCL8* in

**Fig. 1 Establishment of a BAL and blood-derived immune cell atlas from patients with COVID-19. a** Description of the cohort highlighting the source of BAL (only from severe patients, in red) and blood samples (from severe and mild patients and healthy donors, in red, orange and blue, respectively). **b** Two-dimensional UMAP embedding of the scRNA-seq data. Dots (cells) are colored according to their respective metacluster (Epithelial cells, lymphocytes, neutrophils, and monocytes/macrophages). **c** Expression heatmap of the 14 significant clusters detected in our scRNA-seq dataset. The top five best markers were selected for each cluster. **d** Two-dimensional density plot of the UMAP embedding of the BAL (upper panel) or the blood (lower panel) cells. **e** Proportion of neutrophils, lymphocytes, and monocytes/macrophages in blood samples across patient status. For each of the cell types, an ANOVA test was performed and corrected for multiple-testing by Bonferroni correction (one-sided Fisher test). Median and 5–95% theoretical quantiles are shown. $N = 32$ independent clinical samples were used, including 5 derived from healthy patients, 6 from mild patients, and 21 from severe patients. **f** Expression of the *SPP1* (osteopontin) gene across cell types (left panel) and distribution of Macrophages (1) among total BAL cells based on severe patient clinical outcome (right panel). A two-sided Welch's *t* test was performed to compare proportion between the two groups of patients ($t = 2.2$ with a degree of freedom equal to 16.5). Median and 5–95% theoretical quantiles are shown. Normality was tested using a one-sided Shapiro–Wilk test for each individual group ($p = 0.064$ and $p = 0.32$ respectively). $N = 21$ independent clinical samples were used, including 13 derived from patients who survived and 8 from deceased patients.

### Table 1 Anagraphic and coexisting disorders of enrolled patients and healthy donors.

| Characteristics | Healthy controls $N = 5$ | Mild patients $N = 10$ | Severe patients (ICU) $N = 21$ |
|---|---|---|---|
| *Anagraphic* | | | |
| Age, yr: median (IQR)* | 66 (64–73) | 69 (56–80) | 67 (58–70) |
| Male, no. (%) | 4 (80) | 6 (60) | 17 (81) |
| *Coexisting disorder, no. (%)* | | | |
| Any | 2 (40) | 10 (100) | 17 (81) |
| Obesity | 0 (0) | 2 (22) | 3 (14) |
| Hypertension | 2 (40) | 10 (100) | 11 (52) |
| Diabetes | 0 (0) | 3 (30) | 7 (33) |
| Chronic obstructive pulmonary disease | 0 (0) | 2 (20) | 1 (5) |
| Cardiovascular disease | 0 (0) | 5 (50) | 3 (14) |
| Chronic kidney disease | 0 (0) | 2 (20) | 1 (5) |
| Active malignancies | 0 (0) | 0 (0) | 2 (10) |

IQR denotes interquartile range.

cluster (4) and inflammatory response genes (*NFKBIA*, *IL1RN*, and *SOD2*) in cluster (5).

We observed a strong tissue specificity in the cell clusters distribution (Fig. 1c, d). For instance, neutrophil clusters (4) and (5) were BAL specific, whereas clusters (2) and (3) were blood specific, and cluster (1) could be found in both tissues. As expected, M1-like and M2-like macrophages could only be identified in BAL samples while monocytes were mostly blood specific. Epithelial cells were limited to the BAL samples with low representation of the total cell population (1.1% of total cells).

We next analyzed possible associations between the proportion of cell clusters and the clinical status of the patient. We observed that severe patients exhibited a significantly higher proportion of neutrophils in their blood samples compared with mild patients and healthy controls (Fig. 1e, left panel). In contrast, the lymphocyte proportion was decreased in severe patients compared with mild patients and healthy controls (Fig. 1e, middle panel), whereas the monocyte/macrophage compartment was not affected by disease severity (Fig. 1e, right panel). As scRNA-seq is prone to biases for population proportion estimation, we validated our findings by performing blood cell counting and systemically looking for differences between the three groups of patients. Consistently with our scRNA-seq analysis, we observed significant differences in the neutrophil and lymphocyte population, but also a decreased erythrocyte number in severe patients (Supplementary Figure 1f, g). No other cell population was significantly affected by the disease (Supplementary Figure 1f).

COVID-19 is characterized by an excessive inflammatory response that is sometimes referred to "cytokine storm" and is deemed to drive the disease pathogenesis. As inflammatory macrophages are suspected to be the main producer of inflammatory cytokines such as IL-6 and IL-1β, we looked for possible enrichment of inflammatory macrophages in the BAL from patients who died owing to COVID-19. We found that the macrophage (1) cluster had a signature recalling M1-like alveolar macrophages, as previously described[22], characterized by the expression of *SPP1* (osteopontin, Fig. 1f, left panel). Surprisingly, these macrophages were associated with a better prognosis of severe patients (Fig. 1f, right panel), suggesting that this population can be a biomarker of positive outcome. At last, we investigated the concentration of cytokines in the plasma and systemically looked for differential concentration between the classes of patients. Following multiple-testing correction, we observed that only three cytokines were significantly affected by the patient status: vascular endothelium growth factor–alpha, IL-6 and Interleukin -1 receptor antagonist (IL-1RA, encoded by the IL1RN gene). All three exhibited higher concentrations in severe and mild patients' plasma compared with healthy controls, whose levels were close to the detection limit (Supplementary Figure 1h–i).

Altogether, we comprehensively profiled >150,000 immune cells from blood and BAL sampled from COVID-19 patients. Coarse-grained clustering allowed us to detect a severity associated neutrophilia and lymphopenia, but also a SPP1+ macrophage population associated with severe patients' survival. Interestingly, only the concentration of three cytokines was associated with disease severity, suggesting that massive cytokine release in blood is not present in COVID-19 patients. However, the restricted size of our cohort limits the statistical power of our analysis and might hinder the detection of other disease-associated variables.

*COVID-19 is associated with neutrophil activation.* Neutrophils are the most common white blood cells and are the first cells to migrate to the site of infection. Our data set contains 42,238 high-quality blood neutrophils, therefore allowing an in-depth analysis of the association of neutrophil activity and SARS-Cov2 pathogenesis. We performed a refined clustering of neutrophils, which identified 10 different clusters (Fig. 2a, b), including a distinct and rare subtype expressing *CEACAM8*, *LTF*, and *DEFA3* genes corresponding to immature neutrophils[19,20,26]. Among the other clusters, we observed both a resting neutrophil state (*ICAM1*, *CXCL8*) and an array of activated neutrophil clusters. Among them we identified an interferon-stimulated genes (ISGs: *RSAD2*, *OAS2*, *IFIT1*), a serine protease inhibitor (*PI3* and *SLPI*), and a chemokine (*CCL4*, *CCL3L3*) expressing clusters, suggesting of a strong heterogeneity of the neutrophil polarization across patients.

**Table 2 Clinical characteristics of enrolled patients and healthy donors.**

| Characteristics | Healthy controls $N = 5$ | Mild patients $N = 10$ | Severe patients (ICU) $N = 21$ |
|---|---|---|---|
| Median (IQR) interval from symptoms onset (S.O.) and outcome | | | |
| Days from S.O. to Hospitalization | - | 5 (2-6) | 6 (4-7) |
| Days from S.O. to ICU admission | - | - | 7 (6-10) |
| Days from S.O. to dismission from ICU | - | - | 38 (23-45) |
| Days from S.O. to dismission from Verona Hospital | - | 21 (13-24) | 43 (32-66) |
| Outcome, no. deaths (%) | - | 1 (10) | 8 (38) |
| Clinical features at sampling | | | |
| APACHE score, median (IQR) | - | - | 23.5 (15-28.5) |
| SOFA score, median (IQR) | - | 2 (0.8-3.3) | 6 (4-7) |
| pCO2 [35-40 mmHg], median (IQR) | - | 34 (31-41) | 48 (41-52) |
| pO2 [80-100 mmHg], median (IQR) | - | 59 (55-64) | 77 (69-97) |
| FiO2 %, median (IQR) | - | 26 (21-31) | 60 (50-90) |
| P/F ratio mmHg, Median (IQR) | - | 213 (194-267) | 146 (71-177) |
| Score on ordinal scale (1-8), Median (IQR) | - | 4 (3-4) | 7 (7-7) |
| Laboratory findings at sampling, Median (IQR) | | | |
| Hemoglobin (135-160 g/L) | 151 (145-153) | 120 (98-137) | 104 (89-112) |
| Leukocytes (4.3-10 $10^9$/L) | 6.7 (6.2-8.8) | 5.6 (5.2-6.4) | 12.8 (9.6-15.8) |
| Neutrophils (1.8-8 $10^9$/L) | 3.8 (3.4-4.9) | 3.8 (3.1-4.6) | 10.3 (7.7-13.4) |
| Lymphocytes (1.2-4 $10^9$/L) | 2.2 (1.7-2.8) | 1.3 (1-1.8) | 1.2 (0.7-1.4) |
| Monocytes (0.2-1 $10^9$/L) | 0.7 (0.4-0.8) | 0.5 (0.4-0.8) | 0.7 (0.4-1.1) |
| Eosinophils (<0.45 $10^9$/L) | 0.3 (0.3-0.4) | 0.1 (0-0.1) | 0.2 (0.1-0.4) |
| Basophils (<0.2 $10^9$/L) | 0.05 (0.04-0.08) | 0.02 (0.01-0.04) | 0.04 (0.02-0.07) |
| Platelets (150-400 $10^9$/L) | 233 (216-234) | 248 (160-311) | 285 (217-368) |
| C-reactive protein [inf. a 5 mg/L] | - | 24 (22-56) | 131 (75-158) |
| P-Ferritin [30-300 µg/L] | - | 947 (342-1368) | 1262 (735-1647) |
| P-D-Dimer [inf. a 500 µg/L] | - | 1600(690-2501) | 2250 (1319-3684) |
| Thrombin clotting time (pt) 0.82-1.17 INR | - | 1.06 (0.98-1.1) | 1.14 (1.09-1.25) |
| P-fibrinogen [2.00-4.00 g/L] | - | 4.2 (3.3-5) | 6.8 (5.4-7.8) |
| Microbiology analysis on BAL at sampling | | | |
| Lung infection (CFU > $10^4$), no. (%) | - | - | 10 (48) |
| Pseudomonas lung infection (CFU > $10^4$), no. (%) | - | - | 7 (33) |

*IQR denotes interquartile range, PCO₂ carbon dioxide partial pressure, PO₂ oxygen partial pressure, FiO₂ fraction of inspired oxygen, P/F PO₂/FiO₂, CFU denotes colony-forming units.*

To define neutrophil compartment composition in an unsupervised fashion, we used the correspondence analysis (CA), a method similar to principal component analysis but adapted to categorical data (see Methods). CA second component was able to stratify healthy controls, mild, and severe patients (Fig. 2c, Supplementary Figure 2a, b). We observed that severe, and to a lesser extent mild patients, were associated to a replacement of resting neutrophils by multiple clusters including the ISGs, *CD177,* and *PI3*-expressing neutrophils (Fig. 2d, Supplementary Figure 2c). Interestingly, immature neutrophils were only detected in both mild and severe patients, albeit at a low level (<5% of neutrophils)—except in four severe patients (Fig. 2D right panel). We systemically computed Pearson's correlation between CA dimension 2 and each measured biological and clinical variables and identified IL-6 and IL-1RA concentration as the most positively correlated variables, with erythrocyte and partial CO₂ concentration (pCO₂) negatively correlating with CA dimension 2 (Supplementary Figure 2d). Altogether, our refined analysis of blood neutrophils revealed that resting neutrophils are replaced by various neutrophil clusters endowed with inflammatory and immature signatures in both mild and severe patients.

*Myeloid immunesuppression predicts COVID-19 patient survival.* As highlighted by our scRNA-seq analysis, the complexity of the blood neutrophil compartment often characterizes both mild and severe COVID-19 patients, limiting the number of cells other than neutrophils that can be solved by scRNA-seq. Recent evidence suggested the presence of monocyte alteration in SARS-CoV-2-infected patients, mostly associated with the expansion and accumulation of immunosuppressive monocytes[27]. Owing to

the low number of blood monocytes sequenced in our data set (9103 cells, corresponding to <300 cells per patient) and the limited ability of scRNA-seq to provide functional information at this resolution, we purified circulating CD14+ monocytes from blood and used them to perform T-cell immunosuppression assays. Both cellular and supernatant-associated immunesuppressions were assessed. In addition, immunosuppressive activities of both normal density (NDN) and low density (LDN) neutrophil supernatants were also measured.

All samples caused some degree of T-cell suppression, with monocytes and monocyte supernatants exhibiting similar activity, whereas LDN and NDN exhibited the highest and lowest suppression activity, respectively (Supplementary Figure 2e). Suppression rate by monocytes was significantly lower in healthy controls compared with both mild and severe patients but severe patients displayed a higher variance than mild patients, with the suppression rate ranging from 10% to ~80% (Fig. 2e left panel). Surprisingly, this heterogeneity could be partly explained by the clinical outcome of the severe patients: while severe patients who survived displayed high T-cell immunosuppression by monocyte supernatants, monocyte supernatants of deceased patients were unable to dampen T-cell proliferation (Fig. 2e, right panel). This inverse association between immunosuppression and patient survival was not limited to monocyte supernatant as it could be observed, albeit less significantly, with LDN supernatants (Supplementary Figure 2f).

Myeloid cells suppress T-cell activation through multiple strategies including anti-inflammatory cytokine secretion, nutrient depletion, or immune checkpoint engagement[28]. To gain further insight in COVID-19-induced immunesuppression, we

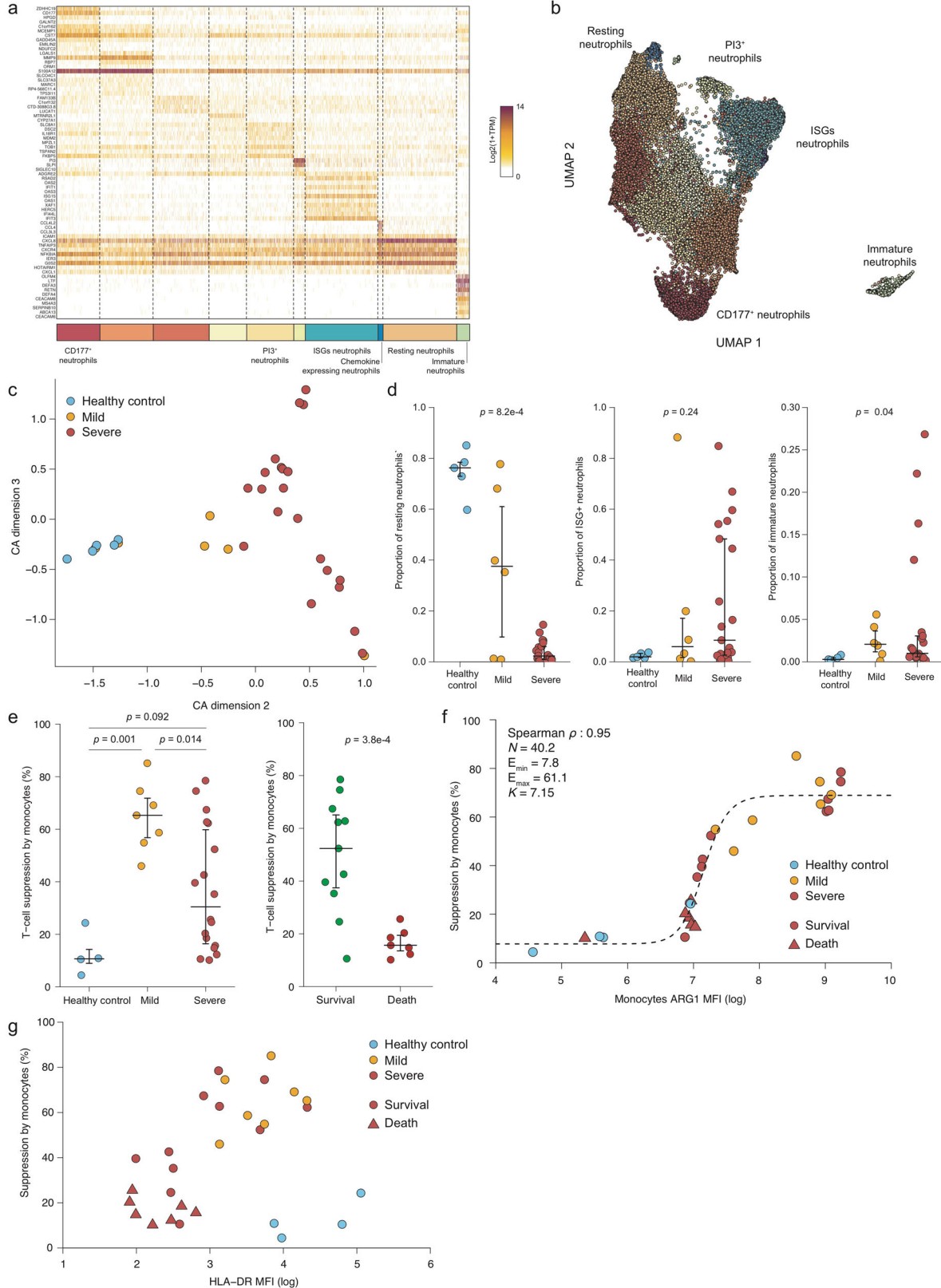

profiled monocyte expression of PD-L1 (CD274), ARG1, and HLA-DR by flow cytometry (Supplementary Figure 2g). We observed a clear relation between mean ARG1 expression by monocytes and monocyte immunosuppressive function (Spearman $\rho = 0.95$; Fig. 2f), which could be fitted using a modified Hill function (see Methods), revealing a strong Hill coefficient

($n = 40.2$). HLA-DR expression was also associated with immunesuppression, but in a different manner compared with ARG1 (Fig. 2g). HLA-DR mean expression and immunesuppressive activity clustered three groups of patients: healthy controls with a high HLA-DR expression and low immunosuppression; mild patients and severe patients who survived with both high

**Fig. 2 Analysis of blood myeloid cells shows unique features associated with patient status and outcome. a** Expression heatmap of the 10 clusters identified among the blood neutrophils. **b** Two-dimensional UMAP embedding of the blood neutrophil. Cells are colored according to their cluster. **c** Scatter plot of the Correspondence Analysis (CA) of the blood neutrophil populations. **d** Proportion of resting neutrophils (left panel), ISGs neutrophils (middle panel) and immature neutrophils (right panel) among blood neutrophils according to patient clinical status. A one-sided Tukey's range test was used in the left panel while Kruskal–Wallis rank test was used in both middle and right panel. Median and 5–95% theoretical quantiles are shown. Normality of the distribution in the left panel was assessed using a Shapiro–Wilk test in each group individually ($p = 0.39$, $p = 0.99$, and $p = 0.043$, respectively). $N = 32$ independent clinical samples were used, including 5 derived from healthy patients, 6 from mild patients, and 21 from severe patients. **e** T-cell suppression ability of CD14$^+$ monocytes according to clinical status (right) or to ICU outcome (severe patients only, right panel). A Tukey's range test was used in the left panel while a two-sided Welch's $t$ test was performed in the right panel ($t = 4.9$ with a degree of freedom equal to 11.8). Median and 5–95% theoretical quantiles are shown. Normality of the distribution in the right panel was assessed using a Shapiro–Wilk test in each group individually ($p=0.91$ and $p=0.7$ respectively). In the left panel, $N = 29$ independent clinical samples were used, including 4 derived from healthy patients, 7 from mild patients and 18 from severe patients. In the right panel, $N = 18$ independent clinical samples were used, including 11 derived from patients who survived and 7 from deceased patients. **f** Association between CD14$^+$ monocytes ARG1 Mean Fluorescence Intensity (MFI) and immunesuppression. The dashed line corresponds to the fitted Hill-like function. $N = 29$ independent clinical samples were used, including 4 derived from healthy patients, 7 from mild patients and 18 from severe patients. **g** Association between CD14$^+$ monocytes HLA-DR Mean Fluorescence Intensity (MFI) and immunesuppression. Source data are provided as a Source Data file.

suppression and high HLA-DR expression; a third group of severe patients with low suppression and HLA-DR expression. As more than half (7/12) of the patients from the last groups died, we hypothesize that this cluster corresponds to patients suffering from terminal immune dysfunctions and therefore at higher risk of fatal outcome. At last, we observed a limited association between PD-L1 and immunosuppression (Spearman $\rho = 0.57$) (Supplementary Figure 2h). Furthermore, the concentration of 20 different cytokines, including both pro-inflammatory (IL-6, TNF) and anti-inflammatory ones (IL-10) was assessed in monocyte supernatant; however, none of the cytokines analyzed correlated with immunesuppression (absolute Spearman $\rho$ lower than 0.4; Supplementary Figure 2i). In summary, the immunosuppressive activity of monocytes and neutrophils is a strong predictor of severe patient survival and is primarily associated with ARG1 expression, and to a lesser extent with PD-L1 but not with any specific cytokine secretion.

*COVID-19 progressively affects blood and lung lymphocyte compartments.* The lymphocyte compartment is extremely heterogeneous and dynamic as it contains various cell types with properties and functions that can evolve upon inflammation and infection. By re-clustering cells identified as lymphocytes, we were able to obtain a finer picture of their heterogeneity. We identified 14 clusters, including several effector and memory T cells, naive T cells, and activated γδ-T cells (Fig. 3a and Supplementary Figure 3a). Interestingly we were able to identify a cluster of B cells expressing high level of TCF4 that was specific to the blood of patient 8, a patient suffering from chronic lymphocytic leukemia (CLL). We assumed that those cells were neoplastic and were thus removed from the analysis. As expected, significant differences could be observed between blood and BAL lymphocytes, with memory, effector, and dividing T cells mostly found in the BAL and NK cells in the blood (Fig. 3a, right panel).

To identify trends in the lymphoid compartment composition in an un-supervised manner, we used CA, as described above. The first CA dimension of the lymphoid population perfectly separated the blood and BAL samples (Fig. 3b and Supplementary Figure 3b) and captured a major trend in the BAL lymphocyte population. We therefore computed the correlation between this dimension and the various clinical features measured and noticed a striking negative correlation with the Sequential Organ Failure Assessment (SOFA) score (Fig. 3c) but not with other variables (Supplementary Figure 3c). In opposition, a parallel analysis performed on the blood of severe patients identified no clinical parameters associated with CA dimension 2 (Supplementary

Figure 3d), strengthening the importance to analyze the main constituency affected by the disease. Among the six major BAL lymphocytes clusters (>5% of total BAL lymphoid cells clusters), CD8$^+$ T resident memory (CD8$^+$ Trm, expressing *ZNF683* and *ITGA1*) had the strongest positive correlation with CA dimension 1 ($R = 0.79$), followed by CD4$^+$ T resident memory (CD4$^+$ Trm) ($R = 0.17$). Interestingly, CD4$^+$ Trm expressed several immune checkpoints such as *CTLA-4* and *PD-1* (encoded by *PDCD1*), suggesting that they may participate in the immune regulatory landscape (Fig. 3d). The naive T-cell cluster (*IL7R*, *LEF1*) had the strongest negative correlation with the CA dimension 1, and therefore the strongest positive association with the SOFA score ($R = -0.66$). Three clusters were negatively associated with CA dimension 1: effector CD8$^+$ T cells ($R = -0.30$, *LAG3* and *CD27*), dividing T cells ($R = -0.16$, *MCM10*, *E2F8*) and activated γδ-T cells ($R = -0.13$, *XCL1/2* and *TRDC*).

To test whether CA was also able to capture meaningful variations in blood lymphocyte population, we analyzed the potential association between patient status and CA dimension 2 as it captured an important trend in blood samples (Fig. 3b). Interestingly, CA dimension 2 was significantly associated with patient clinical status with a higher value being specific for severe patients (Fig. 3e, left panel). This dimension was associated with two populations of NK cells and two populations of γδ-T cells (Fig. 3e, g), i.e., resting/activated NK cells, and resting/activated γδ-T cells. Both activated NK and γδ-T cells were associated with severe COVID-19 patients while resting cells were found in healthy controls and mild patients only (Fig. 3e and Supplementary Figure 3e). Activated NK and γδ-T cells were associated with an increased expression of cytotoxicity genes and activation markers such as *PRF1*, *NKG7*, *KLRG1*, and *CD247* (Fig. 3g) while resting NK cells were featured by a higher expression of the inhibitory KIR receptors *KIR2DL1* and *KIR3DL2* and resting γδ-T cells by a higher expression of *TNF* and *DUSP8* (Fig. 3g). Consistently with our initial analysis of the cytokine and blood cell count data, we found that CA dimension 2 was positively associated with IL-1RA plasma concentration and neutrophil counts and negatively associated with erythrocyte count and hemoglobin concentration (Fig. 3f). Taken together, our analysis of the lymphoid compartment revealed that the presence of a naive T-cell population in BAL is associated with high clinical severity, whereas the blood of severe COVID-19 is characterized by the activation of NK and γδ-T cells.

*Memory T-cell migration and disease severity is linked to CXCR6.* Expression profiles generated by scRNA-seq can be overlaid with genome-wide association studies (GWASs) to pinpoint specific

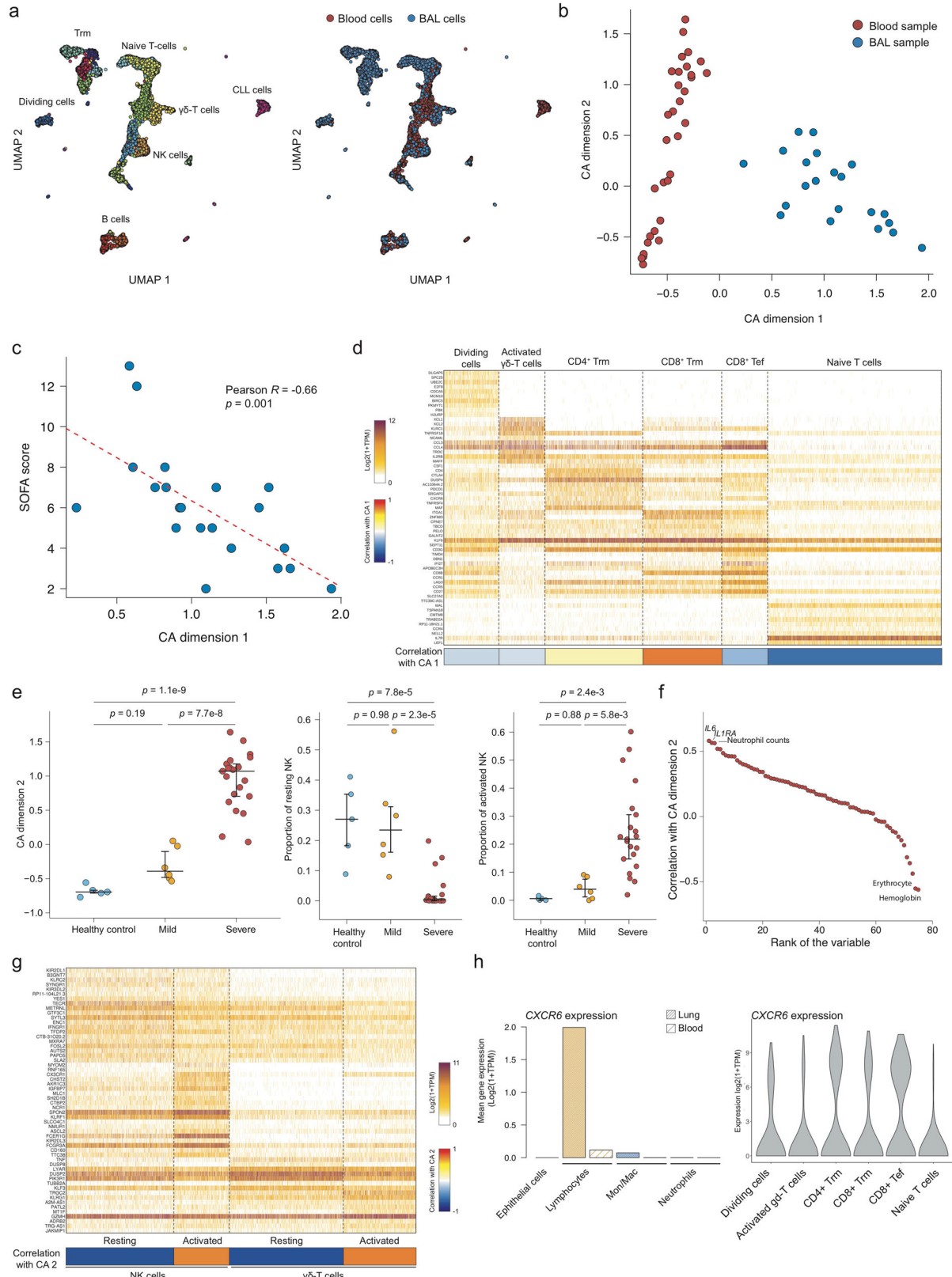

cell types and identify potential cellular and molecular mechanisms, explaining the described genetic associations[29]. In addition to the ABO group locus, a recent study found that another genomic locus is associated with the development of severe forms of COVID-19[23]. However, how this locus contributes to the pathology is unclear. As six different genes were covered by the

peak association (*SLC6A20*, *LZTFL1*, *CCR9*, *FYCO1*, *CXCR6*, and *XCR1*), we looked at their expression in the four different cellular compartments in both blood and BAL samples. Strikingly, only *CXCR6* was expressed at a detectable level (Fig. 3h left panel, S3f) and specifically in BAL lymphocytes, effector, and memory T cells (Tem and Trm, respectively) and not by the naive T-cell cluster

**Fig. 3 Cellular clusters associated with better patient prognosis. a** Two-dimensional UMAP embedding of lymphocytes colored according to their cluster (left panel) or their tissue (right panel). **b** Correspondence Analysis (CA) of blood and BAL lymphocytes. **c** Association between SOFA score and CA dimension 1 score of BAL samples. **d** Expression heatmap of the BAL lymphocytes belonging to the cell clusters that are associated with CA dimension 1. The 10 best markers are shown for each cluster. **e** Distribution of CA dimension 2 score of blood samples according to clinical status (left panel), and proportion of resting (middle panel) and activated NK (right panel) according to clinical status. A one-sided Tukey's range test was used to compute the shown $p$ values. Median and 5–95% theoretical quantiles are shown. $N = 32$ independent clinical samples were used, including 5 derived from healthy patients, 6 from mild patients, and 21 from severe patients. **f** Ranked Pearson correlation between biological features and CA dimension 2. **g** Expression heatmap of the blood lymphocytes belonging to the cell clusters that are associated with CA dimension 2. The 20 best markers are shown for each cluster. **h** Mean expression of *CXCR6* across different tissues and cell types (left panel) and among the different BAL lymphocytes clusters associated with CA dimension 1. Source data are provided as a Source Data file.

(Fig. 3h, right panel). As the risk allele is associated with a decreased expression of *CXCR6* and that *CXCR6* is expressed by key protective populations, our data strongly suggest that patients with the risk allele have a lower amount of the protective T-cell populations in the lung, therefore increasing the risk of developing severe COVID-19 forms.

*Viral landscape of COVID-19 patients affects immune profile.* At the time of ICU admission, most COVID-19 patients exhibited a low viral load in the lung, suggesting that the virus had been mostly eliminated and that at this time the pathology was mainly driven by an inappropriate immune response rather than by viral replication. To validate this hypothesis, we applied our recently published tool, Viral-Track[18] on the BAL-sequencing data in order to quantify the SARS-CoV-2 viral load and detect possible secondary viral infections.

Quantification of SARS-CoV-2 viral reads revealed that in most of severe patients (17/21), no SARS-CoV-2 reads could be found (Fig. 4a). Out of the four SARS-CoV-2-positive patients, three had low levels of viral reads (<10 SARS-CoV-2 reads per million - RPM), whereas patient 8 displayed >1000 SARS-CoV-2 reads RPM. This unusual amount of reads was not due to a bias in the proportion of epithelial cells (Supplementary Figure 4a). Consistent with our previous scRNA-seq study of SARS-CoV-2, we observed that most reads were located in the very 3′-end of the viral genome (Fig. 4b), probably due to the 3′ bias of our scRNA-seq method and to nested replication process of the virus[30]. Surprisingly, we also found that two patients had a significant amount of Herpex Simplex Virus 1 (HSV-1) reads (NCBI reference number: NC_001806) with, respectively, 265.392 and 46.229 viral reads in patients 4 and 25, respectively (Fig. 4c, Supplementary Figure 4b). Coverage analysis revealed dozens of peaks that corresponds to viral genes, suggesting an active transcription of the viral genes (Fig. 4d), whereas the quantification of the viral genes expressed revealed that the most expressed genes include a virion component (*US10*) and inhibitors of the immune response (*US11* and *US12*) (Supplementary Figure 4c). We validated this finding by performing a multiplex PCR test for four different Herpesviridae, HSV-1, HSV-2, human cytomegalovirus (HCMV), and varicella-zoster virus (VZV) on samples from patients 8 (CLL patient, negative control), 13 (negative control), and patient 25 (Supplementary Figure 4). None of the viruses were detected in samples from patient 8 and 13 but HSV-1 was specifically detected in BAL samples of patient 25 that were collected at two different moments during the patients' stay in ICU.

We then looked for a possible explanation for the high SARS-CoV-2 viral load of patient 8. As mentioned before, it appeared that patient 8 suffered from CLL, a B-cell malignancy characterized by the accumulation of small, mature-appearing lymphocytes in the blood, bone marrow, and in lymphatic system[31]. CLL patients often suffer from hypogammaglobulinaemia, i.e., a reduction in maturated and high-affinity antibodies. We therefore quantified the amount of IgG produced toward the receptor-binding domain

(RBD) region of the viral spike protein (Fig. 4e): while healthy controls and most of the mild patients lacked any RBD-targeting IgG, all severe patients but patient 8 had high levels of these immunoglobulins. Therefore, the lack of an efficient antibody response might have prevented the complete clearance of the virus from the lung in this patient. We tried to investigate which BAL cells from patient 8 were infected by SARS-CoV-2. We observed that a reduced number of cells exhibited an extremely high number of viral UMIs (several hundreds to several thousands), but interestingly those cells were mostly apoptotic cells (high levels of mitochondrial reads), neutrophils or in some rare case, cells that expressed extremely low amount of reads (Fig. 4f, Supplementary Figure 4e–f). At last, we looked for the possible effects of high-viral loads on the immune system state: for this purpose, we performed a CA on all blood cell populations. We observed that the first dimension was associated with high-viral load, as patients 4 and 8 had the highest score among all patients with significant correlation between total viral load and the first dimension of the CA ($R = 0.80$, Supplementary Figure 4g). Furthermore, this CA dimension correlated with a subset of blood neutrophils that specifically expressed ISGs such as *IFIT1*, *RSAD2,* and *OAS3* (Fig. 4h, i) but also *PD-L1*, suggesting that a high-viral load in the lung can significantly influence the blood immune landscape in COVID-19 patients.

## Discussion

The real boundaries between sepsis induced by either SARS-CoV-2 or bacteria are still ill-defined and need to be further demarcated. Indeed, hyper-inflammation associated with immunesuppressive state are clinical characteristics shared among these pathologic conditions, which underscore a susceptibility state defined as sepsis-induced immunoparalysis[32]. In this disease phase, innate and adaptive dysfunctions cooperate for the ineffective clearance of the pathogen, vulnerability to secondary infections, and reactivation of latent viruses[33]. Immunoparalysis is likely the ground for the HSV-1 superinfection observed in some ICU patients in our study. HSV-1 infection has been indeed identified in patients receiving prolonged mechanical ventilation and can contribute to worse outcome[34]. A recent analysis confirms HSV-1 reactivation in almost 50% of ARDS COVID-19 patients through longitudinal follow-up, attributed to the immunesuppressed conditions of ARDS COVID-19 patients[35].

In our study, we identified at least 10 different neutrophil states, including resting, activated, and immature cells in whole blood and BAL. Although in mild patients the resting populations were mainly represented, severe patients were characterized by multiple clusters of neutrophils expressing ISG-associated genes, *CD177*, *PI3,* and *CEACAM8*, reminiscent of neutrophil subsets presenting an activated or immature phenotype, in line with recent published data[19,20]. To evaluate the contribution of neutrophils in regulating the immune responses, marked by both immunosuppression and inflammation, we isolated neutrophils from the LDN and NDN fractions from peripheral blood of severe and mild patients. For the first time, we demonstrated at a

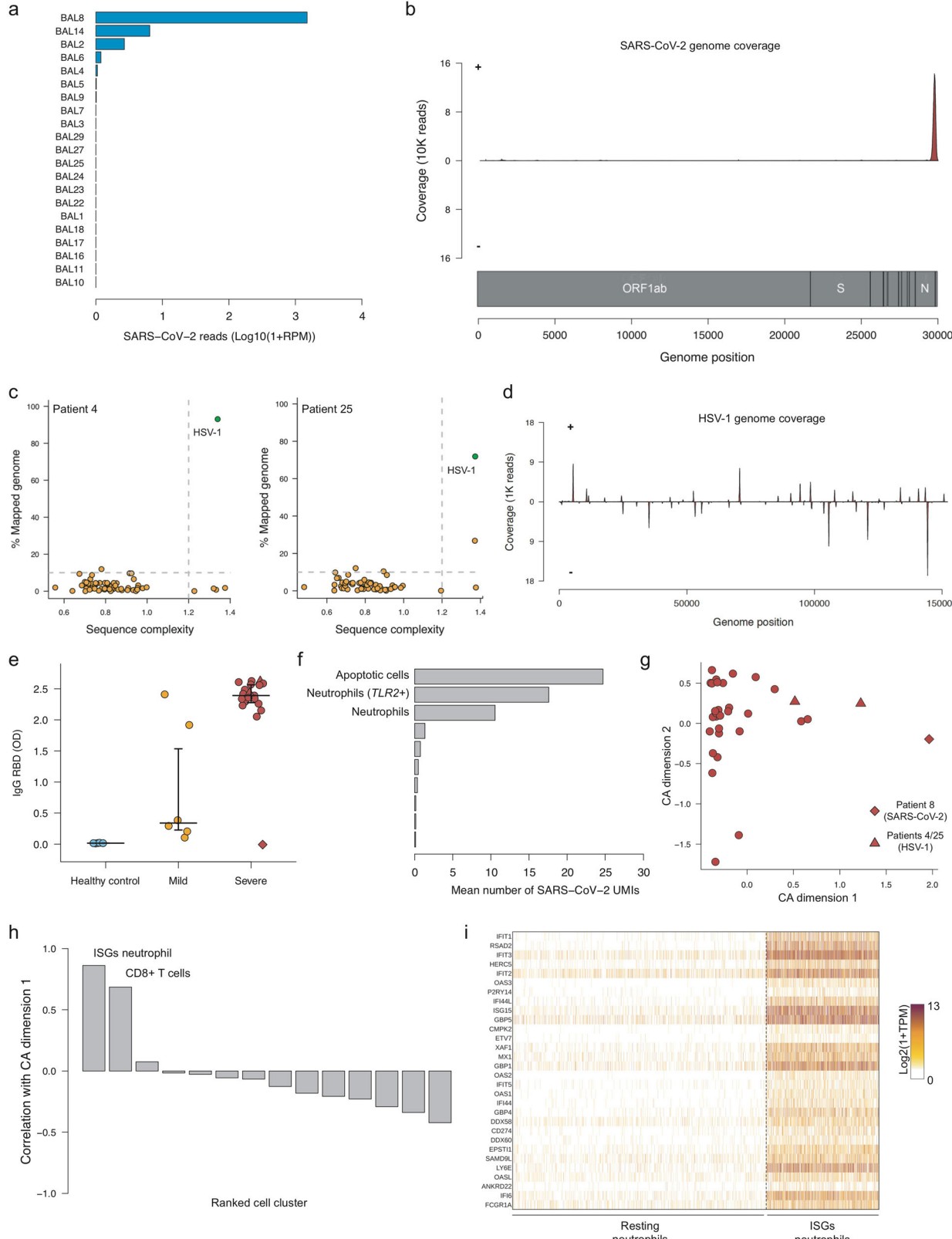

**Fig. 4 Analysis of viral landscape in patients with severe COVID-19. a** Number of SARS-CoV-2 UMIs in each BAL sample. **b** Coverage plot of SARS-CoV-2 genome. **c** Viral-Track analysis of patients 4 and 25. **d** Coverage plot of HSV-1 genome. **e** Quantification of IgG targeting the RBD domain of the SARS-CoV-2's spike protein. OD: optical density. Median and 5–95% theoretical quantiles are shown. $N = 30$ independent clinical samples were used, including 5 derived from healthy patients, 6 from mild patients, and 19 from severe patients. **f** Mean number of SARS-CoV-2 UMIs across patient 8 cell clusters. **g** CA analysis of total blood cell population. **h** Correlation between cell clusters proportion and total blood CA dimension 1. **i** Expression heatmap of cells belonging to resting or ISGs neutrophils in peripheral blood. Source data are provided as a Source Data file.

functional level that LDN neutrophils can suppress T-cell proliferation in COVID-19 patients. More importantly, we identified in the loss of suppressive activity by LDN-derived supernatant a worse clinical outcome biomarker for severe patients (Supplementary Figure 2f, right panel). To support our findings that the aforementioned LDNs are reminiscent of the PMN-MDSCs found in cancer and in sepsis[36,37], we interrogated our scRNA-seq data and found that neutrophils, in both mild and severe COVID-19 patients, contained an immature cluster expressing *CEACAM8* and *DEFA3*, reminiscent of emergency myelopoiesis. Although this is in line with results obtained by Schulte—Schrepping et al.[20], we were not able to observe the upregulation of *CD64* and *PD-L1*, as recently described in neutrophils during sepsis[38] and in neutrophils derived from whole blood of COVID-19 patients[20].

Interestingly, scRNA-seq unveiled a cluster of immature CD14+ monocytes, with low *HLA-DR* and expressing *MPO*, *PLAC8*, and *IL1R2*, in the blood of COVID-19 patients and similar cells were reported in sepsis[20,39]. These findings shed light on the practice to measure the levels of HLA-DR in monocytes as marker for sepsis, preconized but never definitively proven as mortality biomarker for severe sepsis, especially in ICU patients[40,41]. Considering the emerging cell heterogeneity, HLA-DR levels are insufficient to reach a predictive value for patient mortality, if not associated with functional analyses. Indeed, our study shows that HLA-DR reduction in monocytes, an established hallmark of many COVID-19 patients[20], could define severe patients with higher susceptibility to a fatal outcome when combined with the absence of suppressive activity by monocytes (Fig. 2g). CD14+HLA-DR^low monocytes are expanded in severe COVID-19 patients, a likely consequence of the pervasive emergency myelopoiesis triggered by SARS-Cov2 infection[39]. Based on gene expression profiles, these cell subsets were deemed to have immunesuppressive features, which were not functionally addressed in the study. To the best of our knowledge, we show for the first time that immunesuppression is a hallmark of COVID-19 evolution (Fig. 2e) and could be the basis for gradual loss of effector/memory in favor of naive T cells (Fig. 3d). Moreover, we defined a correlation between ARG1 expression and the immunosuppressive activity of monocytes (Fig. 2f), with a minor contribution of PD-L1 (Supplementary Figure 2g). We found that loss of function in myeloid cells mirrors an immune pathological status progressing from immune paralysis to "immune silence" associated with higher susceptibility to death event (Fig. 2e, Supplementary Figure 2f). These results corroborates the previous findings showing the implication of CD14+ARG1+ immune-suppressive monocytes in patients with pancreatic cancer who also share increased levels in some inflammatory cytokines[28,42]. Further studies are necessary to define whether these cells are immature precursors of the granulocytic/monocytic lineage or a new subset of circulating monocytes, as well as the extent of their overlap with the immature CD14+ cells described in COVID-19 patients and sepsis[20,39].

On the light of several recent studies claiming that immuno-suppression is a hallmark of COVID-19[43,44], and support the clinical use of anti-rheumatic drugs, our data pinpoint the repurposing of these drugs exclusively for the treatment of mild and severe patients with a favorable prognosis.

Noteworthy, our study identifies new markers associated with disease severity in the peripheral blood of patients, such as the proportion of activated/resting γδ-T cells (Supplementary Figure 3e), which integrate with others already described and confirmed in our study (i.e., NK cells and neutrophils activation[45,46], which might be responsible for healthy tissue damage[47]). We believe these data might have important implications from a clinical and biological point of views, potentially supporting

physicians in patient stratification. More importantly, the extensive molecular analysis performed in BAL samples reveals that lymphoid cell landscape perfectly mirrors compromised healthy conditions of the patients and identifies in the increase of naive T cells in spite of CD4+ Trm and CD8+ Trm the worst clinical scenario (Fig. 3c, d). These results further expand our knowledge on the alteration of lung T-cell compartment after SARS-CoV-2 infection, as initially described[48].

Naive T cells unbalance is not clearly observed in peripheral blood of the same patients, suggesting that systemic analysis, despite many advantages, will not provide complete picture of the disease. In this study, we found CXCR6 highly expressed in lung CD4+ and CD8+ Trm and Tem, whose accumulation is generally associated with lower SOFA score, as shown in Fig. 3c, d, suggesting that the accumulation of Trm and Tem is beneficial for a better outcome of these patients. Accordingly, *CXCR6* expression was recently associated with less-severe forms of COVID-19[23]. Thus, we envision a scenario in which *CXCR6* orchestrates Trm and Tem cells partitioning within the lung, directing them to the airways[49].

Collectively, the molecular, phenotypic, and functional myeloid and lymphoid cell characterization performed in both peripheral blood and lung points out a profound immune dysregulated status, which can support secondary bacteria and virus infection in critically ill patients (Table 2, Fig. 4c, Supplementary Figure 4a). Indeed, as a sign of immune paralysis, monocytes in septic patients do not respond to LPS stimulation with the upregulation of NF-κB-dependent genes, including TNF[39]. Although this was not investigated in our study, we nonetheless show evidence that monocytes from patients who had a fatal outcome in ICU were dysfunctional and had lost the immune regulatory properties. It is thus likely that monocytes in terminally ill patients are flawed in different biological responses, possibly including the ability to differentiate into M1 macrophages in the lung, as observed in the BAL of these patients. Together with the reset of lymphoid arm indicated by the relative abundance of naive T cells, this configures a state of "immune silence" and supports the deploying of drugs that can "reawaken" host immune system. "Immune silence" could be related to the extensive immaturity of this cell population, as a consequence of an abnormal and skewed myelopoiesis. Alternatively, it might be a pre-existing disorder making this group of patients unable to cope with the hyper-inflammatory state. Longitudinal studies are mandatory to dissect between these different disease developments.

Our data support the administration of drugs that aim at switching off and re-starting the perturbed host immune system in both the adaptive and innate arms, as suggested by the efficacy of recently adopted dexamethasone in severe COVID-19 patients[50]. By combining scRNA-seq with functional assays, we were able to identify features associated with disease severity and clinical outcome of COVID-19. However, our study is not flawless and suffers from limitations, including experimental design ones. First, this is not a longitudinal study. Therefore, it is not possible to track at single-cell level the local and peripheral immune landscape dynamic evolution of patients. Moreover, single sampling approach is very likely underestimating the number of patients facing HSV-1 infection. Second, the absence of BAL samples from mild patients prevents us to generalize and validate the observations made on severe ones. Moreover, the absence of BAL samples from healthy control deprives us of estimating the basal state and composition of the lung immune system. However, the invasiveness of BAL collection procedure and the possible serious negative consequences on the health of both spontaneously breathing, mild COVID-19 patients and healthy donors raises important ethical question that need to be

considered while designing COVID-19 cohorts. Finally, the BAL samples used in our study is composed by a mix of physiologic solution, which is mechanically introduced and bronchoalveolar fluid. This method captures infiltrating immune cells in spite of non-immune cell types (i.e., epithelial cells), as shown by recent autopsy reports[51]. Lung biopsy represents an interesting alternative to BAL but is far riskier, especially for patients suffering from ARDS, such as severe COVID-19 patients, and the amount of available material is limited. An alternative approach would be the collection of samples from deceased patients; however, as most patients died after several weeks in ICU, the biological features observed during the autopsy might either be linked to the disease itself or to the extended stay in ICU. In the case of lung biopsy, highly multiplexed imaging techniques such SeqFISH[52] or CODEX[53], but also spatial transcriptomic technologies[54] could be explored. Such approaches would provide gene expression spatial pattern at a near single-cell resolution and gain essential information concerning potential cellular interactions. At last, we observed that several patients had a significant interferon-induced polarization of blood neutrophils that could not be solely explained by the virus presence in the lung. Such polarizations might also be due to secondary infections, but mostly bacterial rather than viral. Bacterial secondary infection is a condition that critically ill patients often face before entering or during ICU hospitalization. The inflamed fluid-filled alveolar tissue observed during SARS-CoV-2 infection represents indeed an ideal habitat for bacterial growth of pathogens including *Pseudomonas aeruginosa* and *Staphylococcus aureus*[55]. According to this, many studies showed moderate to high percentages (~22–73%) of secondary bacterial infection in critically ill patients hospitalized in ICU during SARS-CoV2 severe pneumonia[3,56,57]. This result strengths and extends what already observed in previous epidemic disease, such as SARS-CoV[58] and MERS[59]. Therefore, an extensive metagenomic analysis of the BAL could reveal disease-associated bacterial landscape and be related to specific immune phenotype.

Overall, our study expands the biological insight on the multifaceted virus–immune system interplay and provides a solid background to design and test new candidate drugs for severe COVID-19 patients.

## Methods

**Study design and clinical considerations**. This study includes a group of 21 severe COVID-19 patients admitted to ICU, 10 mild SARS-CoV-2 patients and 5 healthy donors. The anagraphic and clinical features of the three groups of individuals are recapitulated in Table 1 and 2. The clinical condition of the patients was analyzed according to an eight-category ordinal scale[60]. In brief, score 1 and 2 for not hospitalized patients with no limitations/limitations of activities, respectively; score 3 and 4 for hospitalized patients, not requiring/requiring oxygen therapy by mask, respectively; score 5 and 6 for hospitalized patients requiring non-invasive ventilation/intubation and mechanical ventilation, respectively; score 7 for hospitalized patients requiring multi-organ support other than ventilation; score 8 for death patients. All 31 patients with COVID-19 were admitted, within the period from March 12th to April 20th 2020 to the University Hospital of Verona. At sampling, the stage of disease was categorized as mild (patients not requiring non-invasive/mechanical ventilation and/or admission to ICU, score on ordinal scale 3–4) or severe (patients requiring admission to ICU and/or non-invasive/mechanical ventilation, score on ordinal scale 6–7)[60]. All patients were hospitalized in ICU for respiratory organ failure as proved by their clinical parameters (SOFA score, $pCO_2$, $pO_2$, $FiO_2$, P/F ratio). Within this cohort, one patient was also affected by CLL, two patients displayed also either cardiac or kidney failures. Patients' age range mirrors the characteristics of the individuals admitted in ICU in that time frame and is matched within the three cohorts. Less than 20% of all patients were on steroids at sampling. The study has been designed with the purpose of defining a complete framework of COVID-19 patients' immune landscape. To this aim, we collected clinical (i.e., co-morbidities, pulmonary performances, outcome at dismissal from Verona Hospital) and laboratory (i.e., leukocyte subsets, P-ferritin, P-D-Dimer, C-reactive protein, P-fibrinogen quantification) information and integrated them with molecular (i.e., single-cell transcriptomic analysis) proteomic (cytokines quantification and serology), phenotypic (myeloid

characterization in terms of expression of immunesuppression hallmarks), functional (myeloid immunesuppressive assay) data.

**Ethics approval statement**. All relevant ethical guidelines have been followed, and any necessary IRB and/or ethics committee approvals have been obtained. This study was approved by the Ethics Committee for Clinical Experimentation, Department of Hospital Medical Management, Hospital Trust of Verona in Verona, Italy (protocol 17963 and 51095; principal investigator, Vincenzo Bronte; registered in ClinicalTrials.gov with following id NCT04438629). All participants (and/or initially their families) provided written informed consent before sampling and for the use of their clinical and biological data.

**Preparation of biological samples**. For each severe patient, ~20 ml of BAL fluid was obtained, stored at room temperature and processed within 2 h in a BSL-3 laboratory. No BAL fluid was obtained from mild patients and healthy donors. An unprocessed aliquot was used for bacterial culture. The BAL fluid was filtered two times through a nylon gauze and a 100-μm nylon cell strainer to remove clumps and debris. The supernatant was then washed with PBS 1× and centrifuged. RBCs were lysed with 4 mL of 0.2% NaCl solution (3 min, RT) and the reaction was blocked by adding 9 mL of 1.2% NaCl solution. The cells were washed with PBS 1×, resuspended in Roswell Park Memorial Institute (RPMI) 1640 medium supplemented with 5% bovine serum albumin and counted. Cell viability was determined by Trypan blue exclusion. BAL fluids of patients with COVID-19 infection contained a heterogeneous number of cells ranging from $0.83 \times 10^6$ to $22 \times 10^6$ cells. Cells were resuspended at a concentration of $1 \times 10^6$ /ml for single-cell analysis. Peripheral blood (PB) from COVID-19 patients and healthy donors was collected in EDTA-coated tubes. In all, 2 ml of PB was washed once with PBS 1BAL and the RBCs lysis was performed twice adding 15 mL of 0.2% NaCl solution (3 min, RT) and the reaction was blocked by adding 35 mL of 1.2% NaCl solution. The cells were washed with PBS 1BAL, resuspended in RPMI 1640 medium supplemented with 5% bovine serum albumin, filtered through a 100-μm nylon cell strainer and counted. Cell viability was determined by Trypan blue exclusion. Cells were resuspended at a concentration of $1 \times 10^6$ /ml for single-cell analysis.

**SARS-CoV-2, HSV-1 detection, and bacteria identification**. Nucleic acids were extracted from BAL samples by Seegene Nimbus instrument (Seegene; Seoul, South Korea) and SARS-CoV-2-related genes (E, N, and RdRP) were amplified by polymerase chain reaction (PCR) with the Allplex 2019-nCoV assay kit (Seegene) according to the manufacturer's instructions. BAL samples from COVID-19 positive patients were processed for bacterial isolation. In brief, samples were treated with 1% v-v dithiothreitol for 30 min at 25 °C Upon treatment, (20 μl of) the samples were streaked out on petri dishes containing different types of agar (Blood Agar, Chocolate Agar, Columbia Nalidix Acid agar, Mannitol Salt Agar, McConkey, and Sabouraud Agar). Cultures were incubated for 24 h at 37 °C and bacterial growth was evaluated as colony-forming unit per ml. Bacterial species were identified by MALDI-tof (VITEK-MS, BioMérieux; France) and antimicrobial susceptibility was tested by VITEK-2, BioMérieux; France) and Kirby–Bauer disk diffusion assay. Nucleic acids isolated from BAL samples were processed to multiplex PCR procedure for the simultaneous detection of HSV-1, HSV-2, HCMV, and VZV, using an Allplex assay (Seegene) according to the manufacturer's instructions. Unfortunately, the presence of those viruses could not be assessed in patient 4 due to the limited amount of available BAL fluid.

**Detection of cytokines and serology**. Cytokines released by patients' monocytes and neutrophils were quantified by Human ProcartaPlex™ Panel 1 multiplex (ThermoFisher Scientific, Waltham, MA, USA). The ELISA assay to detect Immunoglobulins (Ig) used fragment of the SARS-CoV2 spike glycoprotein (S-protein) as antigens based on a recently published protocol[61]. The Spike SARS-CoV2 glycoprotein RBD was expressed in mammalian human embryonic kidney cell line 293 (HEK293, ATCC® CRL-1573™, LGC Standards S.r.l., Milano, Italy) at IEO, Milan by Drs. Marina Mapelli and Sebastiano Pasqualato as glycosylated proteins by transient transfection with pGACCS vectors generated in Dr. Krammer's laboratory. Cells were grown in DMEM (Invitrogen, Carlsbad, CA, USA) supplemented with 10 mM 4-(2-hydroxyethyl)-1-piperazineethanesulfonic acid (Euroclone, Milano, Italy), 150 U/ml streptomycin 200 U/ml penicillin (Euroclone, Milano, Italy), 2 mM L-glutamine (Euroclone, Milano, Italy), and 10% heat-inactivated fetal bovine serum (FBS; Invitrogen, Carlsbad, CA, USA). The constructs were synthesized using the genomic sequence of the isolated virus, Wuhan-Hi-1 released in January 2020, and contained codons optimized for expression in mammalian cells. Secreted proteins were purified from the culture medium by affinity chromatography, quantified, and stored in liquid nitrogen in aliquots. The ELISA tests to detect IgG in patients' sera used as antigens the recombinant fragments of the RBD of the Spike SARS-CoV2 glycoprotein. After binding of the proteins to a Nunc Maxisorp ELISA plate, patients' sera to be analyzed were applied to the plate to allow antibody binding, and then revealed with secondary antihuman- IgG (BD) antibody conjugated to enzyme horseradish peroxidase (Thermo Scientific 31410, 0.8 mg/ml, 1:1000). Reaction was revealed upon addition of TMB (Merck). Optical density at 450 nanometers was measured on a Glomax

(Promega) plate reader. All samples were tested and validated with an ELISA assay, as indicated in[62].

**Flow cytometry analysis.** Immunophenotype analysis on whole peripheral blood was performed according to standard procedures in order to characterize monocyte subsets (defined as classical, CD14$^{high}$ CD16$^{low/dim}$; intermediate CD14$^{int}$ CD16$^+$; non classical CD14$^{low/dim}$ CD16$^{high}$). In brief, peripheral blood was incubated with FcR Blocking reagent (Miltenyi Biotec, Paris, FR) followed by the addition of the followings: antihuman PE-conjugated CD56 (BD Bioscience, San Jose, CA, USA; clone NCAM16.2, cat. num. 335791, 25 μl/ml), fluorescein isothiocyanate (FITC)-conjugated-CD16 (BD Bioscience, San Jose, CA, USA; clone 3G8 cat. num. 555406, 20 μl test;), PerCP-Cy5.5-conjugated CD3 (BD Bioscience, San Jose, CA, USA; clone UCHT1, cat. num. 560835, 5 μl test), PE.Cy7-conjugated HLA-DR (eBiosciences, ThermoFisher Scientific, Waltham, MA, USA; clone L243, 335795, 5 μl test), APC.H7-conjugated CD14 (BD Bioscience, San Jose, CA, USA; clone MφP9, cat. num. 560180, 5 μl test), Brilliant Violet 421™-conjugated PD-L1 (BD Bioscience, San Jose, CA, USA; clone MIH1, 563738, 5 μl test;) and alexa fluor 647-conjugated ARG1 (developed in our laboratory[28]) antibodies, Aqua LIVE/DEAD dye (ThermoFisher Scientific, Waltham, MA, USA). RBCs were lysed using Cal-Lyse™ Lysing Solution (ThermoFisher Scientific, Waltham, MA, USA) in accordance with the manufacturer's instructions. Samples were acquired with FACS Canto II (BD, Franklin Lakes, NJ, USA) and analyzed with FlowJo software (Tree Star, Inc., Ashland, OR, USA).

**Myeloid cell isolation, characterization, and functional assay.** Cells were isolated from EDTA-treated tubes (BD Biosciences, NJ, USA) and freshly separated by Ficoll-Hypaque (GE Healthcare, Uppsala, Sweden) gradient centrifugation. PBMCs were counted and the monocyte fraction (CD14$^+$) was further isolated by CD14$^-$ microbeads (Miltenyi), following the manufacturer's instructions. From the CD14$^-$ fraction the CD66$^+$ LDNs were isolated by the sequential addition of CD66b-FITC antibody (BD Biosciences, NJ, USA) and microbeads anti-FITC (Miltenyi), following the manufacturer's instructions. The NDNs CD66b$^+$ were isolated from the RBC layer by dextran density gradient followed by CD66b-antibody as described for LDNs. The purity of each fraction was evaluated by flow cytometry analysis. Samples with a purity >95% were assessed for their suppressive capacity. In all, $0.5 \times 10^6$ cells of each cell type were plated in 24-well plates for 12 h in complete RPMI supplemented with 10% FBS. At the end of the incubation, viability was evaluated by flow cytometry and Trypan blue assay, and both the supernatants and the cells were collected and cultured with CellTrace (Thermo Fisher Scientific) labeled PBMCs, stimulated with coated anti-CD3 (clone OKT-3, eBioscience, Thermo Fisher Scientific 16-0037-81) and soluble anti-CD28 (clone28.2, eBioscience, Thermo Fisher Scientific 16-0289-81) for 4 days in 37 °C and 8% CO$_2$ incubator[63]. For the cells, a ratio of 3:1 (target:effector) was used. At the end of the culture, cells were stained with anti-CD3-PE/Cy7 (UCHT1, BD biosciences 563423, 5 μl/test) and CellTrace signal of lymphocytes was analyzed with FlowJo software v10 (Tree Star, Inc. Ashland).

**Single-cell RNA sequencing.** BAL and peripheral blood cells were isolated and prepared as described above. For each sample, cells were resuspended in RPMI supplemented with 5% FBS to a final concentration of 1000 cells per ml and processed using the 10x Genomics Chromium Controller and the Chromium NextGEM Single-Cell 3′ GEM, Library & Gel Bead kit v3.1 (Pleasanton, California, United States) following the standard manufacturer's instructions. In brief, 10,000 live cells were loaded onto the Chromium controller to recover 4000 single-cell GEMs per inlet uniquely barcoded. After the synthesis of cDNA, sequencing libraries were generated. Final 10× library quality was assessed using the Fragment Analyzer High Sensitivity NGS kit (Agilent Technologies, Santa Clara, CA, USA) and then sequenced on the Illumina NextSeq500 (Illumina, San Diego CA, USA) generating 75 base pair paired-end reads (28 bp read1 and 91 bp read2) at a depth of 50,000 reads/cell.

**Data analysis and statistics**
*Generation of UMI tables.* Upstream processing of reads was done using the CellRanger toolkit with default parameters. SARS-CoV-2 (NCBI reference number: NC_045512.2) and human hg38 genomes were downloaded from NCBI website. SARS-CoV-2 GTF annotation file was downloaded from the UCSC and merged with the human GTF as an additional chromosome. ORF_10 Gene 3′ boundary extended by 100 bases to catch all reads that belong to this transcript.

*High-level analysis of scRNA-seq expression data.* ScRNA-seq expression data analysis were performed using the R-based Pagoda2 pipeline (https://github.com/hms-dbmi/pagoda2/)[24] in addition to an in-house R script. In brief, UMI table were loaded using the read.10x.matrices() function. Low-quality cells were removed using the following strategy: cell with <500 UMIs and >20% of mitochondrial genes were removed. Two rounds of analysis were performed: in the first one, all filtered cells were used to identify the major cell types, then cells from each cellular compartments are analyzed individually to provide more detailed informations. For each analysis, the number of highly variables genes (HVGs) was determined using the adjustVariance() function with the gam parameter set to 10. HVGs were

selected using the following strategy: for each gene, its number of zeros and its mean expression are computed. A local polynomial model is then used to predict the number of zeros according to the log mean expression (loess function with degree parameter set to 2). The residuals of this model (excess of zeros) are then used to ranked the genes and the genes with the highest excess of the zeros are considered as the most HVGs. PCA reduction is then computed using the calculatePcaReduction() function. The number of computed PC was changed in each analysis owing to variable number of cells and cellular heterogeneity. A K-nearest neighbor graph was then build with the function makeKnnGraph() with the K value set to 30 and the distance parameter set to "cosine". In order to get high-quality cell clusters, we used the Leiden community detection implemented in the R package leiden, a wrapper of the python package leidenalg. The leiden() function was applied to the KNN graphs with default parameters for each analysis. Marker genes were identified using the getDifferentialGenes() function. UMAP low-dimensional embedding was computed using the uwot R package, and more precisely the umap() function with the n_neighbors parameter set to 30, and the metric parameter set to "cosine". In order to group clusters of cells in the first round of analysis, mean gene expression of the most variable genes was computed using the aggregate() function. Spearman's correlation matrix was then computed using the cor() function with the method parameter set to "Spearman". Hierarchical clustering was then performed on this matrix using Ward's method and the resulting tree used to aggregate the cell clusters.

*CA of the scRNA-seq data.* In order to identify trends in cellular composition across samples, we used a multivariate technique called CA. CA is highly similar to PCA but is applied to contingency table instead of classical continuous data table. First data are pre-processed by dividing each entry by the sum of all matrix entries resulting in the matrix S. Then, a second matrix is computed by subtracting the expected distribution of samples (obtained by multiplying the row and column marginal probably vectors) resulting in a new matrix M. M is then decomposed using singular value decomposition. Because CA is a descriptive technique, it has the advantage of being applicable to tables whether or not the chi-squared statistic is appropriate. We used the R implementation of CA from the package FactoMineR (CA() function) with default parameters. To determine the significant components, we looked at the scree plot and selected the eigenvalues/component located before the elbow. To improve the quality of our analysis, we removed cell clusters corresponding to red blood cells, platelets, and cancer cells from patient 8.

To detect clinical and biological variables associated with the computed correspondence components we used the following strategy: for cytokine concentrations, we first took the square root of the initial values to get normally distributed variables and then computed Pearson's correlation with each component independently. For the other continuous variables (clinical scores, age, body mass index…), Pearson's correlation was directly computed. To test the association between CA component or a specific cell type proportion and a categorical variable (i.e. patient clinical status and survival) we either applied a Tukey's range test (TukeyHSD() function) if the variable is not heavy-tailed. If the cell proportions are clearly heavy-tailed, we applied a Kruskal–Wallis rank test. Normality of the variables was checked using the Shapiro–Wilk test, through the R function shapiro.test().

*Viral-Track analysis.* To detect and study viruses in our scRNA-seq samples, we used Viral-Track, a computational tool that screen the raw sequencing files to find viral reads (32479746). Processing of the file was performed using UMI-tool (28100584). First, cell barcodes were extracted and a putative whitelist computed using the umi_tools whitelist command with the parameters '–stdin —bc-pattern = CCCCCCCCCCCCCCCCCNNNNNNNNNN –log2stderr'. Following the mapping of the reads to viral genomes and transcript assembly, the mapped reads were assigned to transcripts using the R package Rsubread through the function featureCounts() with default parameters. The command "umi_tools count" is then used to compute the final expression table with the following parameters:–per-gene–gene-tag = XT–assigned-status-tag = XS–per-cell.

In the case of patient 8, cells were not filtered on total host UMIs and proportion of MT UMIs but only on total combined host and viral UMIs to avoid removing apoptotic cells containing a high-viral load but expressing few host genes.

*Quantification of HSV-1 gene expression.* Transcriptome annotation file for the NC_001806 viral segment was manually downloaded from the NCBI server. BAM files containing the HSV-1 reads from patients 4 and 25 were loaded into R using the GenomicAlignments package and gene expression quantification done using the featureCounts() function from the Rsubread package with default parameters.

*Analysis of the serum cytokine, blood cell count, and clinical data.* Using a Cullen and Frey graph (descdist() function from the fitdistrplus package), we observed that both serum cytokine and blood cell count variables could be transformed into gaussian-like variables by applying a simple square root function and then used for further analysis. Association between blood cell counts or serum cytokine concentration and patient clinical status was assessed by fitting an analysis of variance (ANOVA) model to the transformed variables (aov() and anova() functions). Correction for multiple testing was done using the p.adjust function with

parameter method set to 'BH'. When correlations with a CA dimension were computed, the cor() function with default parameters was used. To validate the association between the SOFA score and the lymphoid CA dimension 1 we fitted a basic linear model with the lm() function and assessed the significance of the association by performing a Fisher test with the anova() function.

*Analysis of the immunosuppression, flow cytometry, and cytokine secretion data.* As both flow cytometry and cytokine secretion data were extremely heavy-tailed we applied a logarithmic transformation with a pseudo count of 1 ($\log10(1 + x)$). Spearman correlations between protein MFI or cytokine concentration and immunesuppression was computed using the cor() function.

In order to model the relationship between ARG1 MFI and immunesuppression, we applied a function similar to the Hill function used in biochemistry and to model drug dose-response curves:

$$S = E_{\min} + \frac{E_{\max}}{1 + \left(\frac{K}{x}\right)^n} \tag{1}$$

Here $S$ corresponds to the immunesuppression, $x$ to the transformed ARG1 MFI, $E_{\min}$ to the basal immunesuppression, $E_{\max}$ to the maximal suppression that can be induced by ARG1, $K$ to the transformed ARG1 MFI required to get half of the maximal suppression ($E_{\min} + E_{\max}$) and $n$ the cooperation coefficient. This function was fitted using the nls() function with default parameters.

Quantitative variables indicated in Tables 1 and 2 were expressed as the median and interquartile range (IQR), qualitative ones as percentages. All statistical analyses were performed using R 3.6.1 on an Ubuntu 18.04 workstation.

**Reporting summary**. Further information on research design is available in the Nature Research Reporting Summary linked to this article.

## Data availability
The scRNA-seq data have been deposited in the Gene Expression Omnibus (GEO) under accession code GSE157344. The authors declare that all other data supporting the findings of this study are available within the article and its supplementary information files. Source data are provided with this paper.

## Code availability
All scripts used for the data analysis are available at https://github.com/PierreBSC/Verona_COVID19.

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

## Acknowledgements

This work was supported by Fondazione Cariverona (ENACT Project) and Fondazione TIM. We thank all patients who participated in this study and their families. We thank all the members of Immunology Section of Verona University Hospital who actively worked during the pandemia: Francesca Hofer, Varvara Petrova, Chiara Musiu, Cristina Frusteri, Veronica Batani, Alessandro Ghirardelli, Morena Martini, Fiorenza Paiola, Elena Lucchini, Claudia Pizzoli, Elena Chiesa, Oretta Gabrielli, Nadia Brutti, Monica Brentegani, Elisabetta Gallo, Giulio Fracasso, Tiziana Cestari, Ornella Poffe, Antonio Vella, Giovanna Zanoni, Silvia Sartoris, Riccardo Ortolani, Selena Gomirato, Daniel Lovato, Antonella Valentini and Claudia Italia. We thank the members of Division of Infectious Diseases who actively assist patients: Monica Brentegani, Alessandro Visentin, Amina Zaffagnini, and Leonardo Motta. We thank the members of ICU who were actively involved in patients care and assistance: Sara Boschetti, Francesca Del Favero, Gaia Pavan, Riccardo Boetti. We thank Marina Mapelli and Sebastiano Pasqualato of European Institute of Oncology for technical support. We thank 10× Genomics Inc. and Carlo Erba to support the research. We deeply acknowledge the contribution of 'Centro Piattaforme Tecnologiche" of the University of Verona for samples sequencing. I.A. is an Eden and Steven Romick Professorial Chair, supported by Merck KGaA, Darmstadt, Germany, the Chan Zuckerberg Initiative (CZI), the ISF Israel Precision Medicine Program (IPMP) 607/20 grant - P128245, the HHMI International Scholar award, the European Research Council Consolidator Grant (ERC-COG) 724471-HemTree2.0, an SCA award of the Wolfson Foundation and Family Charitable Trust, the Thompson Family Foundation, an MRA Established Investigator Award (509044), the Israel Science Foundation (703/15), the Ernest and Bonnie Beutler Research Program for Excellence in Genomic Medicine, the Helen and Martin Kimmel award for innovative investigation, the NeuroMac DFG/Transregional Collaborative Research Center Grant, an International Progressive MS Alliance/NMSS PA-1604 08459, Adelis Foundation grant, The Wolfson Family Charitable Trust & The Wolfson Foundation, ISF (grant No. 3652/19) within the KillCorona – Curbing Coronavirus Research Program, and Miel de Botton.

## Author contributions

P.B. designed and developed Viral-Track, performed computational and statistical analysis and wrote the manuscript. F.D.S. performed sample processing, bio banking and single-cell experiments, coordinated the study, data analysis, and wrote the manuscript. S.Can. performed sample processing, bio banking, functional studies, and data analysis. S.U. coordinated the study and wrote the manuscript; K.D. was responsible for biological specimens and patients' data collection and revised the manuscript; M.C. performed single-cell experiments; D.E. contributed to data processing and analysis; A.F., C.A., R.M.B., and R.T. performed sample processing, bio banking, functional studies, and data analysis; S.Cal contributed to computational analysis; A.L., M.I., and F.F. performed cytokine and serological analysis and revised the manuscript; D.G. are A.R.M. were responsible for microbiological analysis and revised the manuscript; P.D.N., E.T., L.G., and E.P. were responsible for biological specimens and patients' data collection and revised the manuscript; B.S. contributed to the development of computational methods and bioinformatic analysis and revised the manuscript; I.A. and V.B. directed the project, conceived, designed the experiments, and wrote the manuscript.

## Competing interests

The authors declare no competing interests.
