## [Peer Review File · Nature Communications]

Parts of this Peer Review File have been redacted as indicated to remove third-party material where no permission to publish could be obtained.

REVIEWER COMMENTS

Reviewer #2 (Remarks to the Author):

The authors have reduced some of my concerns through argumentation, and toned down some strong statements

Reviewer #3 (Remarks to the Author):

We thank the authors for incorporating many of our comments into their revised manuscript. We believe this dataset represents an important contribution to the scientific community that warrants publication. Below, we respond to aspects of the manuscript that we believe merit additional attention/consideration.

C. Data & methodology: validity of approach, quality of data, quality of presentation

Major comments:

For maximum reproducibility and generalizability of these findings, it would be helpful to present the patients not as "mild" and "severe" but rather by categories of their WHO ordinal scores. Are the "mild" patients really "mild" (as they are hospitalized)? Are they hospitalized for comorbid medical conditions (like patient 8)? If so, this should be discussed. What is the cause of death for the deceased "mild" patient?

Author's reply. We thank the reviewer for her/his comment. Analyzed patients were stratified in mild and severe groups not only according to clinical (SOFA score) and respiratory parameters, but also according to the adoption of an invasive/mechanical ventilation. All patients were admitted to ICU for respiratory organ failure. Within this cohort, one patient was also affected by CLL, two patients displayed either cardiac or kidney failures. The respiratory parameters (pCO₂, pO₂, FiO₂ %, P/F ratio) and clinical features presented in this work, provide a quantitative evaluation of the clinical condition of the patients. Finally, the patient stratification in mild and severe groups was driven according to these clinical parameters broadly accepted by scientific community and employed in similar COVID-19 studies published in high impact journals^{15 16}. In order to make this message clearer to the reader, we added a better description of the patient clinical state in the revised manuscript.

We thank the reviewer to highlight his/her concern about the mild death patient. Indeed, the deceased patient was a 93 years old individual who entered the emergency hospital after a stroke. The patient was admitted at the hospital again 9 days later after COVID-19 diagnosis. The patient died 5 days later of cardio-respiratory arrest, probably due to a multiple factor combination (mainly age, neurological disorders and COVID-19).

RE-review reply:

We thank the authors for including more clinical parameters in their manuscript, but still feel that the WHO ordinal score would help to clear up confusion. It is true that many papers have used such descriptive terms early in the epidemic, but this has led to confusion about 'mild' patients in the ICU vs. mild as outpatient, as an example. WHO scoring would significantly maximize generalizability, particularly as other groups will no doubt be very interested in this important dataset. Specifically, the labeling of one group of patients as "mild" could be misleading to some readers as these patients were hospitalized and likely fall into the WHO's "moderate" category of score 3-5.

D. Appropriate use of statistics and treatment of uncertainties

While use of ANOVA in Fig 1e is appropriate, it is not the most informative. Comparisons between each group should be performed.

Welch's t-test in Fig 1f is inappropriate as it assumes normality and no test for normality has been performed.

Author's reply. We thank the reviewer for her/his comment. We agree that checking the normality of the sample is required before performing a Welch's test as suggested by the reviewer. By applying a Shapiro-Wilk normality test on the different subgroups of patients, it appears that Welch's test can be applied to figure 1f ($p=0.064$ and $p=0.32$).

Re-Review reply: Thank you for assessing normality. It would be helpful to include the results of the Shapiro-Wilk test in the corresponding figure legends and mention this procedure in the methods.

Tukey's test as applied in Fig 2d-e, 3e is inappropriate as it assumes normality and no test for normality has been performed.

Author's reply. We agree with the reviewer that checking the normality of the sample is essential before performing a Tukey's test. By applying a Shapiro-Wilk normality test on the different subgroups of patients, it appears that Tukey's test can be applied in both figure 2d ($p=0.39$, $p=0.99$ and $p=0.043$) and figure 2e ($p=0.91$ and $p=0.76$).

Re-review reply: Thank you for assessing normality. It would be helpful to include the results of the Shapiro-Wilk test in the corresponding figure legends and mention this procedure in the methods.

Several conclusions are not statistically supported (eg. lines 157-162) and need to be confirmed using appropriate differential expression testing.

Re-review reply: This comment was not addressed by the authors. We now refer to what is line 153 in the revised manuscript.

Comment on why the use of a Hill-like equation (or references for doing so) was performed for Fig 2f, especially when using the Hill's coefficient as a marker for statistical power. Further explanation for what the inflection point we are seeing means in the context of the data. Is there a case for binary classification of suppression states?

Author's reply. We agree with the reviewer's point. Our objective was to determine which protein was essential in the immune suppression triggered by the monocytes. Therefore, we fitted a specific mathematical model to the two candidate genes measured (PD-L1 and ARG1). We assumed that the relationship between the suppressive ability and the gene expression must be monotonically increasing and saturating for a sufficient level of protein expression. Hill-functions fulfill such criterion, are flexible and are thoroughly used to model various biological processes¹⁸. While the ARG1 based model was able to explain most of the variance of the immune suppression by monocytes ($R^2=0.92$), the PD-L1 based model explained less than half of the variance ($R^2=0.39$). Using this modeling approach, we therefore conclude that ARG1 is more likely to be the main mediator of monocyte-induced T-cell suppression.

Re-review reply. It would be helpful to mention in the manuscript the assumptions that went into this application--ie that the suppressive capacity should be monotonic and saturating, and whether it is necessarily sigmoidal. Is there evidence that cooperativity should be modeled for this phenotype?

Why is a different pipeline for Viral-Track Analysis used for sample 8 (preservation of potentially

apoptotic cells to optimize for viral reads)? Might this method potentially pull out some other viruses in other samples if used for other samples?

Author's reply. We thank the reviewer to highlight this possible issue. Viral-Track pipeline was applied as described in our previous paper: we only modified the filtered out cells in our single-cell analysis pipeline as we did not want to remove infected cells (the filtering is performed on the host UMIs). We confirmed that this change does not affect the global ability of Viral-Track to detect viruses.

Re-review reply. Thank you for the explanation. We are referring to lines 706-708 in the revised manuscript. Specifically, why is patient 8 treated differently for Viral-Track analysis?

E. Conclusions: robustness, validity, reliability

The only rationale for the discussed connection between the rs11385942 risk allele and CXCR6 is that the other genes covered by peak association are not detectable in this dataset. This argument does not rule out contributions from the other loci. While the posited hypothesis in the main text is appropriately tempered, it does not seem to warrant an abstract-level claim. There are important implications with identifying a "risk allele." The hypothesis "patients with the risk allele have a lower amount of protective T cell populations in the lung" can be further substantiated using public data to better support this statement.

Author's reply. We thank the reviewer for her/his comment. We are grateful to the reviewer for pointing out this possible issue. We agree that our claim is way to strong and that our computational strategy is flawless. However, it is important to note that the populations pointed out by our analysis are strongly associated with COVID-19 disease severity, therefore supporting our strategy. In order to improve our analysis, we have re-analyzed the published dataset from Vieira Braga et al. (A cellular census of human lungs identifies novel cell states in health and in asthma, GEO identifier: GSE130148) using Pagoda2 pipeline. This dataset consists in more than 10.000 cells sequenced from lung resection using the Drop-seq approach. The lung samples derived from deceased donors with unknown disease. Out of the 6 candidate genes, only one was detected (more than 50 UMIs in the total dataset), FYCO1. While being detected, FYCO1 was only expressed at very low level (mean $\text{Log}_2(1+\text{TPM})$: 0.23) and had unspecific expression pattern across the 20 different clusters identified. We hypothesize that CXCR6 was not detected due to the low amount of T cells sequenced (~8%) and because those cells were mostly naive T-cells and not memory T cells. This second analysis therefore supports the idea that the candidate genes are not expressed by lung non-immune cells. We make this point clearer in the revised manuscript

Re-review reply. We thank the authors for the new analysis, but remain concerned that detection of CXCR6 does not necessarily mean that it is the causal source of the risk allele. Lack of detection of other neighboring genes in lung cells also does not imply the risk allele is related to non-lung cells. We appreciate the better explanation in the manuscript itself, but still feel the claim is rather strongly posited in the abstract.

The authors detect HSV-1 in BAL of 2 COVID-19 patients, but only validate the infection in one donor. Without validation of both patients with this finding, the claim of HSV-1 superinfection as an opportunistic virus in COVID-19 patients also seems inappropriate as an abstract-level claim. This is confusing given the authors' previous description of human metapneumovirus superinfection in COVID-19 using the same in silico strategy ([https://www.cell.com/cell/pdf/S0092-8674\(20\)30568-7.pdf](https://www.cell.com/cell/pdf/S0092-8674(20)30568-7.pdf)), which was also reported without appropriate orthogonal validation. At most, we believe the authors can conclude that viral superinfections may exist in a subset of COVID-19 patients. Also, can the conclusion be made that COVID-19 could be the opportunistic virus in active HSV-1 cases and/or that this is solely a co-infection between both viruses? The data present makes causation claims difficult.

Author's reply. We thank the reviewer to point out these issues. In agreement with these comments, we modified the text by moderating our conclusion and adding several citations to discuss the herpes virus reactivation in ICU patients¹, including COVID-19 patients².

Re-review reply. We thank the authors for removing HSV-1 superinfection as an abstract-level claim. However, we still feel the concluding sentence of the abstract ("immune paralysis that supports secondary bacteria and virus infection") may require tempering, as there is little data directly supporting the connection between the immunophenotypes described and the potential superinfections.

F. Suggested improvements: experiments, data for possible revision

What is the basis for the annotation of low density neutrophil (LDNs) in Fig 2a? No citations are provided. Based on the heatmap, this cell population appears extremely similar to immature or developing neutrophils reported previously (Wilk, et al. Nat Med, Schulte-Schrepping et al. medRxiv, and Aschenbrenner, et al. medRxiv); could consider citing all and considering relabeling.

Author's reply. We thank the reviewer for pointing out this potential issue. We agree that LDNs have been assimilated to immature neutrophils and this should be taken into account in the manuscript and the citations added. The key marker used to define the LDN was CEACAM8, known as CD66b that has been shown to be key marker of LDN 19. This reference was added to the revised manuscript.

We thank the authors for changing the nomenclature to developing neutrophils throughout the manuscript. However, we do not agree that CD66b is a marker of LDNs. The cited manuscript shows higher expression of CD66b by LDNs in HIV, but LDNs are still universally positive for CD66b. Indeed, CD66b is expressed by all mature neutrophils, and it is established that this is one example of CEACAM8 transcript not correlating with CD66b expression: immature neutrophils express CEACAM8 but not CD66b, mature neutrophils express CD66b with little detectable CEACAM8. The expression of multiple early granule markers (LTF, DEFA3, DEFA4) strongly indicate these cells are not LDN or LDN-like but rather immature neutrophils similar to those reported in Wilk, et al. Nat Med and Schulte-Schrepping, et al. Cell.

We are grateful to the Reviewers for their positive assessment and constructive suggestions throughout the review process. We appreciate that the Reviewers recognize the significance and impact of our work. We are mostly grateful for their important and thoughtful comments throughout the manuscript revision, which improved the impact and clarity of the manuscript overall. In the revised manuscript and point-by-point response below, we address all comments raised by the Reviewers.

Referee #1 (Remarks to the Author):

Major comments:

While we believe it is acceptable to include Patient 8 in the analysis as it is individually labeled when relevant, we believe it is still necessary that the authors provide more information about individual-level contributions to the analysis. n is better than many published studies but highly imbalance between the patient severity groups. We request the authors include the following information in a supplemental table or figure:

- A supplementary table of the absolute cell numbers from each patient in each cell type.
- A UMAP projection of the entire dataset colored by patient of origin.

Author's reply. We thank the Reviewer for these important points. Following the Reviewer advice, we added in the revised manuscript a supplementary table (supplementary table 1 in the revised manuscript) including absolute cell numbers from each patient for each relevant cell type.

Healthy Controls	Epithelial cells	$\gamma\delta$ -T cells	NK cells	CD4+ T cells	CD8+ T cells	B cells	Mono cytes	Macro phages (1)	Macro phages (2)	Neutro phils (1)	Neutro phils (2)	Neutro phils (3)	Neutro phils (4)	Neutro phils (5)
Blood_38	2	219	696	191	1	426	230	11	0	21	564	2	10	29
Blood_39	2	357	194	88	0	138	227	20	0	37	1357	8	43	34
Blood_40	1	474	491	89	4	125	344	19	0	62	1067	3	27	53
Blood_41	1	176	138	24	0	60	109	3	0	12	402	28	11	18
Blood_42	1	447	69	35	6	63	204	15	0	25	605	15	4	6
Mild Patients	Epithelial cells	$\gamma\delta$ -T cells	NK cells	CD4+ T cells	CD8+ T cells	B cells	Mono cytes	Macro phages (1)	Macro phages (2)	Neutro phils (1)	Neutro phils (2)	Neutro phils (3)	Neutro phils (4)	Neutro phils (5)
Blood_32	0	51	31	47	1	2	418	4	0	85	1472	95	47	0
Blood_33	0	133	173	49	4	117	275	1	0	7	217	2653	18	11
Blood_34	1	175	121	307	13	86	232	10	1	23	268	36	3	0
Blood_35	1	150	55	78	7	42	333	4	0	37	1285	132	47	20
Blood_36	0	249	525	72	6	36	1058	5	0	301	13	4	3	1
Blood_37	0	150	208	85	8	91	241	5	0	54	140	5	8	5
Severe Patients	Epithelial cells	$\gamma\delta$ -T cells	NK cells	CD4+ T cells	CD8+ T cells	B cells	Mono cytes	Macro phages (1)	Macro phages (2)	Neutro phils (1)	Neutro phils (2)	Neutro phils (3)	Neutro phils (4)	Neutro phils (5)
BAL_01	11	0	0	1	0	0	12	273	0	8	218	62	434	293
BAL_02	182	131	135	255	796	2	0	2139	5	95	14	4	178	4
BAL_03	5	3	0	1	6	0	1	186	2	37	72	20	1853	184
BAL_04	72	6	18	20	63	3	0	168	0	144	3	2	35	27
BAL_05	5	0	6	1	22	0	2	406	2	83	73	3	1128	249
BAL_06	92	12	3	56	33	0	3	200	1	41	28	7	365	21
BAL_07	79	1	1	8	10	2	11	907	0	35	12	39	2053	1260
BAL_08	14	3	0	6	51	0	5	97	0	807	37	2	347	13
BAL_09	59	9	5	74	177	1	14	765	104	333	324	25	1382	107
BAL_10	7	15	0	40	164	1	10	215	1541	50	123	7	768	95
BAL_11	19	2	6	2	16	0	2	345	1	75	39	1	598	180
BAL_14	208	5	18	22	55	5	10	449	2	220	54	3	558	94
BAL_16	2	6	0	11	10	2	4	583	3	31	35	0	1669	440
BAL_17	10	2	3	17	9	1	4	742	6	39	349	52	2907	141
BAL_18	12	0	0	0	3	0	8	133	0	14	108	32	174	2783
BAL_22	161	12	17	101	171	4	1	173	76	910	180	8	2237	18
BAL_23	210	63	48	1201	1085	38	2	161	2012	48	38	2	185	262
BAL_24	111	1	0	4	27	0	14	389	15	607	128	15	1182	101
BAL_25	46	22	14	34	29	6	16	298	10	256	40	6	760	8
BAL_27	218	14	10	50	43	1	5	196	5	115	101	13	614	44
BAL_29	20	7	3	6	10	0	2	220	6	435	2	1	1601	84
Blood_01	0	32	44	18	1	130	190	4	0	9	1185	242	10	80
Blood_02	0	58	24	27	6	55	270	0	0	154	649	186	3	0
Blood_03	0	6	18	9	0	41	139	2	0	65	1138	412	13	9
Blood_04	0	179	124	36	61	70	347	2	0	1782	251	53	1	8
Blood_05	0	32	33	15	9	26	161	1	0	120	420	51	5	3
Blood_06	0	5	9	37	0	41	80	0	0	17	116	21	2	2
Blood_07	0	52	99	10	0	5	122	1	1	5	554	267	15	1
Blood_08	2	144	7	78	30	1	159	0	0	2009	613	341	3	1
Blood_09	0	52	16	108	0	29	253	3	0	53	321	62	5	0
Blood_10	1	141	36	143	0	154	462	5	0	100	2145	114	4	3
Blood_11	0	12	44	48	1	12	257	5	0	476	297	116	8	11
Blood_14	0	41	62	58	2	17	188	4	0	126	43	8	0	1
Blood_16	2	398	134	29	4	99	331	3	0	15	717	221	8	1
Blood_17	0	1	38	6	0	25	260	0	0	96	1356	230	1	1
Blood_18	0	37	107	27	1	42	325	2	0	6	1411	908	2	5
Blood_22	0	21	28	35	0	20	136	4	0	16	936	60	2	0
Blood_23	0	56	70	84	0	51	515	7	0	20	1083	123	29	30
Blood_24	0	34	24	12	2	24	243	1	0	610	463	102	2	0
Blood_25	0	83	82	21	1	40	206	0	0	300	128	37	0	0
Blood_27	4	26	78	24	0	26	191	0	0	98	1739	223	5	0
Blood_29	0	25	19	8	0	5	112	1	0	131	206	741	8	4

Supplementary table 1: Absolute cell numbers of immune cells identified in each patient's specimen referred to Figure 1C.

Moreover, we also performed a UMAP projection of the entire dataset colored by patient of origin, included in the PBP response. Given the high number of samples, we believe that a supplementary table can be more informative and for this reason, we did not include the UMAP as supplementary figure.

Point by point reply Figure 1: UMAP projection of the entire dataset colored by patient of origin. Each dot color represents a patient.

We request the authors to consider this point further. While the proportions of peripheral monocytes do not change in COVID-19, this does not mean there is lesser perturbation. In fact, in multiple single-cell transcriptomic studies of COVID-19, peripheral monocytes are among the most strongly reconfigured cell types. While the authors state in their response that they are not as perturbed, this is not shown in the manuscript.

Author's reply. We thank the Reviewer for raising this important point. Indeed, we analyzed the circulating monocyte population but decided not to highlight these results for the following reasons:

- First, the monocytes are relatively rare, limiting the statistical significance of the analysis. As previously stated, only a median number of 277 circulating monocytes per patient can be identified, preventing any robust and in-depth analysis.
- Second, our analysis did not reveal any consistent 'structure' of monocytes across our patient group in the data. Like for the other cell population we performed unsupervised clustering followed by Correspondence Analysis (CA). While the two first components corresponded to donor-specific populations, the third dimension was able to separate the donors into two groups, control patients and severe patients, with mild patients in-between. However, we could not identify any clinical variables correlating with the dimension 3 (no correlation bigger than 0.25) and dimension 4 could not be considered as significant (a clear elbow can be observed in the Scree plot).

Since, monocytes have been extensively described in various papers and our analysis did not highlight any robust results, we decided not to focus on the monocyte subset in this manuscript for clarity reasons, we believe the reviewer will agree with this decision.

Point by point reply Figure 2: Correspondence Analysis (CA) of the circulating monocyte population. Scatter plot of the CA dimension in which each dot corresponds to a unique donor/patient (red: severe patients, orange: mild patients, grey: healthy controls). Proportion of the variance explained by each dimension (right).

While we agree that this manuscript presents the first functional evidence for the suppressive capacity of LDNs, the authors' reading of Schulte-Schrepping et al. does not fully capture that study. In that paper, neutrophils are sequenced from whole blood in addition to the PBMC fraction. Thus, further discussion specifically comparing findings in the neutrophil compartment from Schulte-Schrepping et al. is merited.

Author's reply. We thank the Reviewer for her/his comment and we apologize for our typo on the Schulte-Schrepping's data. In the revised discussion of our manuscript, we thoroughly compared our findings on the neutrophils with the data published by Schulte-Schrepping and co-authors.

It would be much more clear if the fact that the BAL samples only come from ICU patients was stressed earlier in the manuscript. The first mention of this reads "we performed scRNA-seq analysis on BAL and matched peripheral blood samples." While this is technically true, it can easily be construed that the peripheral blood samples are matched to the BAL. Please change this text and label on the figure itself that BAL samples are only from severe patients.

Author's reply. We thank the Reviewer for highlighting this point. Following the Reviewer's suggestion, we specified in the text and in figure 1a and respective legend the source of BAL samples.

Thank you for the explanation, but the connection between lung viral load and peripheral immunophenotype is not evident from this heatmap. First, it is not mentioned in the figure or the legend if the heatmap is from BAL or blood. Second, to show that viral load in lung is connected with phenotype in blood, the authors should include a metadata vector on the heatmap of the patient from which each cell came from and the viral load in that patient.

Author's reply. We thank the Reviewer for her/his suggestions. In the revised manuscript we modified Figure 4 legend to clarify that the Correspondence Analysis originates from data of blood and not of BAL samples. In addition, to analyze the association between the lung viral load and the

blood immune profile we computed the correlation between the total viral load (sum of the SARS-CoV-2 and HSV-1 $\log_{10}(1+TPM)$ levels) and observed a significant positive correlation ($R=0.80$, $p= 1.86e-08$). We have added this analysis to supplementary figure 4 of the revised manuscript. We added a panel (Figure 4h) in the revised manuscript to better describe the correlation between ISG neutrophil and CA dimension 1.

Point by point reply Figure 3: Association between total viral load in the lung and blood CA dimension 1. Correlation of different leukocyte subsets with CA dimension 1 in peripheral blood.

Reviewer #2 (Remarks to the Author):

The authors have reduced some of my concerns through argumentation, and toned down some strong statements

Author's reply. We thank the Reviewer for his/her positive comments.

Reviewer #3 (Remarks to the Author):

We thank the authors for including more clinical parameters in their manuscript, but still feel that the WHO ordinal score would help to clear up confusion. It is true that many papers have used such descriptive terms early in the epidemic, but this has led to confusion about ‘mild’ patients in the

ICU vs. mild as outpatient, as an example. WHO scoring would significantly maximize generalizability, particularly as other groups will no doubt be very interested in this important dataset. Specifically, the labeling of one group of patients as “mild” could be misleading to some readers as these patients were hospitalized and likely fall into the WHO’s “moderate” category of score 3-5.

Author’s reply.

[Redacted]

Point by point reply Figure 4: Ordinal scale for assessment of COVID-19 disease stage.

Thank you for assessing normality. It would be helpful to include the results of the Shapiro-Wilk test in the corresponding figure legends and mention this procedure in the methods.

Author’s reply. We thank the Reviewer for her/his suggestion. In the revised manuscript we have added the corresponding p-values into the figure legends and the method section.

Thank you for assessing normality. It would be helpful to include the results of the Shapiro-Wilk test in the corresponding figure legends and mention this procedure in the methods.

Author's reply. We thank the Reviewer for her/his suggestion. In the revised manuscript we have added the corresponding p-values into the figure legends and the method section.

Several conclusions are not statistically supported (e.g. lines 157-162) and need to be confirmed using appropriate differential expression testing. This comment was not addressed by the authors. We now refer to what is line 153 in the revised manuscript.

Author's reply. Following the Reviewer's suggestion, we created an Excel file containing the gene expression levels for the 5 different neutrophil clusters and added these data in a new supplementary table (Supplementary table 2) of the revised manuscript.

It would be helpful to mention in the manuscript the assumptions that went into this application--ie that the suppressive capacity should be monotonic and saturating, and whether it is necessarily sigmoidal. Is there evidence that cooperativity should be modeled for this phenotype?

Author's re-reply. We thank the Reviewer for raising this point. According to the data, a sigmoidal distribution of the points is assumed. Thus, we hypothesized that a Hill-function would better describe the association between the two parameters. Therefore, the use of this function is supported by the following observations and assumptions:

- **Monotonicity of the curve:** we made the simple assumption that the more cells express immuno-suppressive proteins, the more immuno-suppressive they are. Therefore, the function used to model the association between immune-suppression and protein expression has to be monotonically increasing.
- **Saturation of the curve:** the suppressive ability of the cells/supernatant was computed as one minus the observed ratio of T-cell proliferation between stimulated T-cells with suppressive cells or supernatant and stimulated T-cell alone. As the proliferation of the T-cells will be higher without supernatant/cells, this suppression ability will be always between 0 and 1 and is therefore bounded. Combined with the monotonicity of the curve, the use of a function saturating at 0 and 1 is therefore necessary.
- **Sigmoidal shape of the curve and potential cooperativity:** it is well established that Arginase (ARG1) like many enzymes works as a trimer², therefore validating the need of using sigmoidal function (i.e. Hill coefficient strictly bigger than one).

In addition, Hill function is generally used to model dose-response relationship, therefore validating our use of such function to model the link between ARG1/PD-L1 expression and immune suppression.

Thank you for the explanation. We are referring to lines 706-708 in the revised manuscript. Specifically, why is patient 8 treated differently for Viral-Track analysis?

Author's reply. Patient 8 was the only patient with a significant number of SARS-CoV-2 reads, and therefore the only one for which we could perform a single-cell analysis of the viral infection. As previously stated, we did not change the way Viral-Track analysis was run but only how we filtered out cells: in the case of patient 8, we observed that when applying the filter of 500 host UMIs, most of the viral reads were removed. This is in line with recent reports suggesting of host mRNA degradation associated to SARS-CoV-2 infection

- <https://www.biorxiv.org/content/10.1101/2020.11.25.398578v1>. We therefore decided to keep cells even if they exhibit relatively low total host UMI count.

E. Conclusions: robustness, validity, reliability

We thank the authors for the new analysis, but remain concerned that detection of CXCR6 does not necessarily mean that it is the causal source of the risk allele. Lack of detection of other neighboring genes in lung cells also does not imply the risk allele is related to non-lung cells. We appreciate the better explanation in the manuscript itself, but still feel the claim is rather strongly posited in the abstract.

Author's reply. We followed Reviewer's advice and modified the abstract accordingly. Indeed, in the revised manuscript we removed from the abstract all the sentences referring to CXCR6 as a risk-allele.

We thank the authors for removing HSV-1 superinfection as an abstract-level claim. However, we still feel the concluding sentence of the abstract ("immune paralysis that supports secondary bacteria and virus infection") may require tempering, as there is little data directly supporting the connection between the immunophenotypes described and the potential superinfections.

Author's reply. We followed Reviewer's advice and modified the abstract accordingly by removing any connection of altered immune functions with secondary bacteria and virus infection.

F. Suggested improvements: experiments, data for possible revision

We thank the authors for changing the nomenclature to developing neutrophils throughout the manuscript. However, we do not agree that CD66b is a marker of LDNs. The cited manuscript shows higher expression of CD66b by LDNs in HIV, but LDNs are still universally positive for CD66b. Indeed, CD66b is expressed by all mature neutrophils, and it is established that this is one example of CEACAM8 transcript not correlating with CD66b expression: immature neutrophils express CEACAM8 but not CD66b, mature neutrophils express CD66b with little detectable CEACAM8. The expression of multiple early granule markers (LTF, DEFA3, DEFA4) strongly indicate these cells are not LDN or LDN-like but rather immature neutrophils similar to those reported in Wilk, et al. *Nat Med* and Schulte-Schrepping, et al. *Cell*.

Author's reply. We thank the Reviewer for her/his comment. We apologize for the misunderstanding in the previous Reply to the Reviewer in which we referred CEACAM8 as CD66b. Indeed, to avoid possible misleading messages to the reader and in agreement with the open-access data by Schulte-Schrepping et al and Wilk et al., we defined immature neutrophils as CEACAM8-, LTF- and DEFA3-expressing cells.

References:

1. Beigel JH, et al. Remdesivir for the Treatment of Covid-19 - Final Report. *N Engl J Med* **383**, 1813-1826 (2020).
2. Caldwell RW, Rodriguez PC, Toque HA, Narayanan SP, Caldwell RB. Arginase: A Multifaceted Enzyme Important in Health and Disease. *Physiol Rev* **98**, 641-665 (2018).

REVIEWERS' COMMENTS

Reviewer #3 (Remarks to the Author):

The authors have responded to our remaining concerns.